# RankSEG-RMA: An Efficient Segmentation Algorithm via Reciprocal Moment Approximation

**Zixun Wang**
Department of Statistics and Data Science
The Chinese University of Hong Kong
1155225012@link.cuhk.edu.hk

**Ben Dai**
Department of Statistics and Data Science
The Chinese University of Hong Kong
bendai@cuhk.edu.hk

## Abstract

Semantic segmentation labels each pixel in an image with its corresponding class, and is typically evaluated using the Intersection over Union (IoU) and Dice metrics to quantify the overlap between predicted and ground-truth segmentation masks. In the literature, most existing methods estimate pixel-wise class probabilities, then apply argmax or thresholding to obtain the final prediction. These methods have been shown to generally lead to inconsistent or suboptimal results, as they do not directly maximize segmentation metrics. To address this issue, a novel consistent segmentation framework, RankSEG, has been proposed, which includes RankDice and RankIoU specifically designed to optimize the Dice and IoU metrics, respectively. Although RankSEG almost guarantees improved performance, it suffers from two major drawbacks. First, it is its computational expense—RankDice has a complexity of $\mathcal{O}(d \log d)$ with a substantial constant factor (where $d$ represents the number of pixels), while RankIoU exhibits even higher complexity $\mathcal{O}(d^2)$, thus limiting its practical application. For instance, in LiTS, prediction with RankSEG takes 16.33 seconds compared to just 0.01 seconds with the argmax rule. Second, RankSEG is only applicable to overlapping segmentation settings, where multiple classes can occupy the same pixel, which contrasts with standard benchmarks that typically assume non-overlapping segmentation. In this paper, we overcome these two drawbacks via a *reciprocal moment approximation* (RMA) of RankSEG with the following contributions: (i) we improve RankSEG using RMA, namely RankSEG-RMA, reduces the complexity of both algorithms to $\mathcal{O}(d)$ while maintaining comparable performance; (ii) inspired by RMA, we develop a pixel-wise score function that allows efficient implementation for non-overlapping segmentation settings. We illustrate the effectiveness of our method across various datasets and state-of-the-art models. The code of our method is available in: https://github.com/ZixunWang/RankSEG-RMA.

## 1 Introduction

Semantic segmentation is a fundamental task in computer vision that assigns each pixel in an image to a specific class, serving as a cornerstone for applications such as autonomous driving [Cordts et al., 2016, Feng et al., 2020], medical image analysis [Heller et al., 2019, Bilic et al., 2023], and augmented reality [Ko and Lee, 2020].

Evaluating the performance of segmentation models naturally requires appropriate metrics that accurately reflect segmentation quality. Specifically, pixel-wise accuracy (Acc) is often biased toward classes that occupy large image regions and fails to account for false positives [Everingham et al., 2010, Wang et al., 2023a]. Consequently, the Intersection over Union (IoU) and Dice metrics have emerged as the standard evaluation measures for semantic segmentation [Cordts et al., 2016,

Zhou et al., 2017]. However, regardless of the metrics employed, most existing works adhere to a classification-based segmentation procedure: (i) training-step: training a model to estimate pixel-wise class probabilities using a strictly proper loss [Gneiting and Raftery, 2007] (e.g., cross-entropy loss [Mao et al., 2023]); (ii) prediction-step: followed by applying argmax or thresholding to these probabilities for the final prediction [Chen et al., 2017, Zhao et al., 2017, Xie et al., 2021]. Yet, as demonstrated by Dai and Li [2023], the prediction-step by argmax and thresholding are inconsistent, meaning that even with an infinite number of data and perfect probability estimation, those approaches still cannot achieve optimal performance in terms of IoU and Dice metrics. Therefore, these methods are typically suboptimal in practical applications.

An alternative direction is designing surrogate loss functions which attempt to optimize IoU or Dice directly, with the most popular approaches being soft-IoU/Dice loss [Rahman and Wang, 2016, Sudre et al., 2017, Eelbode et al., 2020] and Lovász extension loss [Yu and Blaschko, 2018, Berman et al., 2018]. However, Lovász hinge loss has been shown to be inconsistent by Finocchiaro et al. [2022], and consequently, its empirical performance improvements remain controversial [Ma et al., 2021, Dai and Li, 2023]. For soft-IoU/Dice loss, the consistency remains unclear. Nevertheless, soft IoU/Dice loss functions are non-convex, making optimization challenging and unstable in practice. Perhaps for this reason, soft-IoU/Dice loss is typically used in combination with cross-entropy through ad hoc training strategies, with final segmentation predictions made using argmax and thresholding operations. These approaches generally require tuning an additional hyperparameter—the weight between cross-entropy and soft-IoU/Dice loss, resulting in high computational costs and inconvenience in practice.

To this end, a ranking-based consistent segmentation rule (RankSEG; Dai and Li [2023]) is specifically developed to directly optimize IoU and Dice metrics. Unlike the argmax rule and surrogate loss functions, RankSEG offers provable consistency and practical performance improvement. Furthermore, compared to surrogate loss functions, RankSEG only modifies the prediction-step and can serve as a *plug-and-play* module by directly utilizing a model trained with cross-entropy loss, simply replacing the argmax operation in prediction-step.

While theoretically sound, their approach exhibits notable limitations: (1) the algorithms are computationally intensive for high dimensional data—with RankDice, the less demanding of the two, having a time complexity of $\mathcal{O}(d \log d)$ with a large constant factor, where $d$ is the number of pixels. For example, it requires 16.33 seconds on the LiTS [Bilic et al., 2023] dataset, compared to only 0.01 seconds by the argmax rule. (2) In multiclass segmentation, the algorithms are only applicable in overlapping settings where multiple classes can occupy the same pixel, which deviates from standard benchmarks [Everingham et al., 2010, Cordts et al., 2016], and also restricts the application of RankSEG in certain scenarios, such as panoptic segmentation [Kirillov et al., 2019].

**Contribution.** In this paper, we leverage **reciprocal moment approximation (RMA)** in segmentation to address the aforementioned disadvantages with the following contributions:

- We propose RankSEG-RMA, which reduces the computational complexity of RankSEG (both IoU and Dice) to $\mathcal{O}(d)$ while preserving comparable performance.

- We develop a pixel-wise score function based on RMA, enabling efficient adaptation to non-overlapping segmentation settings, in line with standard benchmarks.

- We have theoretically established the quality of the proposed RMA (Theorem 2), and empirical evidence demonstrate that our method not only outperforms the conventional argmax rule but also significantly reduces computational costs compared to existing RankSEG algorithms.

## 2 Background

In this section, we begin by distinguishing between two different definitions of the IoU and Dice metrics: $\text{IoU}^\text{D}/\text{Dice}^\text{D}$ and $\text{IoU}^\text{I}/\text{Dice}^\text{I}$ [Wang et al., 2023a], advocating for the latter in practical applications. Building upon $\text{IoU}^\text{I}/\text{Dice}^\text{I}$, we review RankSEG and its approximated algorithms.

For clarity, our discussion starts with binary segmentation, with extensions to multiclass segmentation presented in Section 3.2. Let $\mathbf{X} \in \mathbb{R}^d$, $\mathbf{Y} \in \{0, 1\}^d$ represent the random variables for an image and its corresponding segmentation mask, respectively. Consider a dataset $\{(\mathbf{x}_i, \mathbf{y}_i)\}_{i=1}^n$ consisting of $n$ realizations. The segmentation function $\boldsymbol{\delta} : \mathbb{R}^d \to \{0, 1\}^d$ produces a predicted mask $\boldsymbol{\delta}(\mathbf{x}) \in \{0, 1\}^d$

for a test image $\mathbf{x} \in \mathbb{R}^d$. We denote $p_j(\mathbf{x}) = \mathbb{P}(Y_j = 1|\mathbf{x})$ as the conditional probability of pixel $j$ being a foreground pixel given the image $\mathbf{x}$. Index set $\{1, \cdots, d\}$ is denoted as $[d]$.

## 2.1 Dice/IoU metrics and its variations in implementation

Dice/IoU metrics are defined based on true positives (TP), false positives (FP), and false negatives (FN). However, in practical implementations, the calculation of these components (TP, FP, FN) can be specifically defined at either the dataset or image level, yielding two different metric implementations. For example, the dataset-level and image-level TP are computed as follows:

$$\mathrm{TP}^{\mathrm{D}}(\boldsymbol{\delta}) = \left(\mathbf{y}_1^{\intercal}, \cdots, \mathbf{y}_n^{\intercal}\right)^{\intercal} \left(\boldsymbol{\delta}^{\intercal}(\mathbf{x}_1), \cdots, \boldsymbol{\delta}^{\intercal}(\mathbf{x}_n)\right) = \sum_{i=1}^n \mathbf{y}_i^{\intercal} \boldsymbol{\delta}(\mathbf{x}_i), \quad \mathrm{TP}_i^{\mathrm{I}} = \mathbf{y}_i^{\intercal} \boldsymbol{\delta}(\mathbf{x}_i).$$

Specifically, dataset-level TP aggregates values across the entire dataset, while image-level TP is computed separately for each image. This distinction leads to different averaging strategies when calculating Dice and IoU metrics. Furthermore, dataset-level and image-level Dice are defined as:

$$\mathrm{Dice}^{\mathrm{D}}(\boldsymbol{\delta}) = \frac{2\mathrm{TP}^{\mathrm{D}}(\boldsymbol{\delta})}{\mathrm{TP}^{\mathrm{D}}(\boldsymbol{\delta}) + \mathrm{FP}^{\mathrm{D}}(\boldsymbol{\delta}) + \mathrm{FN}^{\mathrm{D}}(\boldsymbol{\delta})}, \quad \mathrm{Dice}^{\mathrm{I}}(\boldsymbol{\delta}) = \frac{1}{n} \sum_{i=1}^n \frac{2\mathrm{TP}_i^{\mathrm{I}}(\boldsymbol{\delta})}{\mathrm{TP}_i^{\mathrm{I}}(\boldsymbol{\delta}) + \mathrm{FP}_i^{\mathrm{I}}(\boldsymbol{\delta}) + \mathrm{FN}_i^{\mathrm{I}}(\boldsymbol{\delta})},$$

where $\mathrm{FP}^{\mathrm{D}}, \mathrm{FP}^{\mathrm{I}}$ and $\mathrm{FN}^{\mathrm{D}}, \mathrm{FN}^{\mathrm{I}}$ are defined analogously at the dataset-level or image-level.

Although $\mathrm{IoU}^{\mathrm{D}}/\mathrm{Dice}^{\mathrm{D}}$ are more prevalent in the literature [Everingham et al., 2010, Cordts et al., 2016], a growing trend [Liu et al., 2023, Kirillov et al., 2023, Wang et al., 2023a] recognizes $\mathrm{IoU}^{\mathrm{I}}/\mathrm{Dice}^{\mathrm{I}}$ as more favorable for two key reasons. Firstly, $\mathrm{IoU}^{\mathrm{D}}/\mathrm{Dice}^{\mathrm{D}}$ exhibit a bias toward large objects [Yang et al., 2022], which dominate the confusion matrix. This is particularly concerning given the size imbalance in existing datasets [Wang et al., 2023a]. In safe-critical applications, such as medical image analysis or autonomous driving, failing to detect small but critical objects can be catastrophic. Secondly, $\mathrm{IoU}^{\mathrm{I}}/\mathrm{Dice}^{\mathrm{I}}$ offers statistical information at the image-level. For instance, the variance of $\mathrm{IoU}^{\mathrm{I}}/\mathrm{Dice}^{\mathrm{I}}$ quantifies robustness, and the lower quantile measures worst-case performance [Wang et al., 2023a]. Consequently, we adopt $\mathrm{IoU}^{\mathrm{I}}/\mathrm{Dice}^{\mathrm{I}}$ as the focus in this paper. Notably, these two types of metrics differ significantly at the population level. RankSEG-based methods are designed to optimize image-level metrics, which in turn may consequently result in decreased performance on dataset-level metrics.

## 2.2 RankSEG and its blind approximation

For simplicity, we will omit the dependence on $\mathbf{x}$ hereafter, but it is important to note that all following notations are conditional on $\mathbf{X} = \mathbf{x}$. RankSEG [Dai and Li, 2023] establishes a novel segmentation framework that directly (or consistently) maximizes Dice/IoU metrics. Specifically, it first ranks the pixel-wise class probabilities and then selects the top $\tau^*$ pixels as segmented pixels, where $\tau^*$ is so-called the optimal volume. This framework is primarily motivated by the optimal rule outlined in the following theorem; a similar result for $\mathrm{IoU}^{\mathrm{I}}$ is omitted for brevity.

**Theorem 1** (The Bayes rule for $\mathrm{Dice}^{\mathrm{I}}$-segmentation [Dai and Li, 2023]). *Assume that $Y_i \perp Y_j|\mathbf{X}$. A segmentation rule $\boldsymbol{\delta}^*$ is a global maximizer of $\mathbb{E}(\mathrm{Dice}^{\mathrm{I}}(\boldsymbol{\delta}))$ if and only if $\delta_j^* = \mathbb{1}(p_j \geq p_{j_{\tau^*}})$, where $j_\tau$ is the index with the $\tau$-th largest probability. The optimal volume $\tau^*$ is given by:*

$$\tau^* = \underset{\tau \in \{0,1,\cdots,d\}}{\mathrm{argmax}} \ \pi(\mathcal{J}_\tau) \quad with \quad \pi(\mathcal{J}_\tau) = \sum_{j \in \mathcal{J}_\tau} \mathbb{E}\left(\frac{2p_j}{\tau + \Gamma_{-j} + 1}\right), \tag{1}$$

*where $\mathcal{J}_\tau = \{j' : p_{j'} \geq p_{j_\tau}\}$ is the index set of the top $\tau$ conditional probabilities with $\mathcal{J}_0 = \emptyset$, and $\Gamma_{-j} = \sum_{j' \neq j} B_{j'}$ is a Poisson-binomial random variable with $B_{j'}$ being a Bernoulli random variable with success probability $p_{j'}$.*

An intuitive interpretation of Theorem 1 is that $p_{j_{\tau^*}}$ serves as an adaptive threshold that varies across different input images, in contrast to the fixed threshold (0.5) commonly used in binary segmentation framework. This adaptation, in return, indicates that a fixed threshold framework leads to suboptimal performance in terms of $\mathrm{Dice}^{\mathrm{I}}$. This is illustrated by the following example.

**Example.** *Consider $d = 2$ with $p_1 = 0.7, p_2 = 0.4$. The Bayes rule produces $\boldsymbol{\delta}^* = (1,1)^{\intercal}$, whereas the conventional thresholding or argmax rule yields $\widetilde{\boldsymbol{\delta}} = (1,0)^{\intercal}$. Since $\mathrm{Dice}^I((1,1)^{\intercal}) \approx 0.827 > 0.607 \approx \mathrm{Dice}^I((1,0)^{\intercal})$, the thresholding or argmax rule is suboptimal.*

**Blind approximation (BA).**  The primary computational bottleneck in RankDice is the optimization of the optimal volume. Specifically, computing $\pi(\mathcal{J}_\tau)$ for all $\tau \in \{0, 1, \cdots, d\}$ in (1) has a complexity of $\mathcal{O}(d^2)$. To mitigate this, Dai and Li [2023] proposed *RankDice-BA*, which replaces $\Gamma_{-j}$ with $\Gamma$ to make the expectation independent with index $j$, yielding an approximation for $\pi(\mathcal{J}_\tau)$:

$$\pi_{\text{BA}}(\mathcal{J}_\tau) = \mathbb{E}\left(\frac{2}{\tau + \Gamma + 1}\right)\left(\sum_{j \in \mathcal{J}_\tau} p_j\right) = \left(\sum_{l=0}^{d} \frac{2\mathbb{P}(\Gamma = l)}{\tau + l + 1}\right)\left(\sum_{j \in \mathcal{J}_\tau} p_j\right). \tag{2}$$

Fast Fourier transform (FFT) is then used to reduce the overall complexity in evaluating $\pi_{\text{BA}}(\mathcal{J}_\tau)$ for all $\tau \in \{0, 1, \cdots, d\}$ to $\mathcal{O}(d \log d)$. While this achieves a significant improvement, BA method still exhibits the following limitations: (1) the constant factor associated with FFT is generally non-negligible in practice; (2) it is challenging to apply in non-overlapping settings, as shown in Dai and Li [2023, Lemma 7]; and (3) BA is not readily applicable to RankIoU due to the large approximation errors, which therefore remains $\mathcal{O}(d^2)$ time complexity. To address these limitations, we propose a reciprocal moment approximation that further reduces the complexity of both RankDice and RankIoU to $\mathcal{O}(d)$ and enables efficient solution for non-overlapping segmentation.

# 3 RankSEG-RMA

## 3.1 Reciprocal moment approximation

We begin by introducing the reciprocal moment approximation, which is a technique for approximating the reciprocal moment (or negative first moment) of a Poisson-binomial random variable.

**Theorem 2** (Reciprocal moment approximation to RankSEG). *Let $\Gamma$ be a Poisson-binomial random variable, then for any $\tau \geq 1$, we have*

$$(\mathbb{E}\Gamma + \tau)^{-1} \leq \mathbb{E}(\Gamma + \tau)^{-1} \leq \left(\frac{d+1}{d}\mathbb{E}\Gamma + \tau - 1\right)^{-1}. \tag{3}$$

*Therefore, we propose the following $\pi_{RMA}(\mathcal{J}_\tau)$ to approximate $\pi(\mathcal{J}_\tau)$ in (1):*

$$\pi_{RMA}(\mathcal{J}_\tau) = \frac{2}{\tau + \mathbb{E}\Gamma + 1}\left(\sum_{j \in \mathcal{J}_\tau} p_j\right), \tag{4}$$

*and its approximation error for any set $\mathcal{I} \subseteq [d]$ and $\tau = |\mathcal{I}|$ is bounded by:*

$$|\pi_{RMA}(\mathcal{I}) - \pi(\mathcal{I})| \leq 2(\mathbb{E}\Gamma + \tau)^{-1}. \tag{5}$$

Theorem 2 provides two main results: (i) the RMA approximation form (4) for approximating $\pi(\mathcal{J}_\tau)$, inspired by the exchange of expectation and reciprocal in (3); and (ii) a provable error bound that characterize the quality of the RMA approximation. The primary advantage of using RMA is that it avoids expanding the reciprocal moment (RM) into a sum of $d$ terms, which is computationally expensive. Specifically, $\pi_{\text{RMA}}(\mathcal{J}_\tau)$ converts such a nonlinear expectation into a linear one, allowing the evaluation of $\pi_{\text{RMA}}(\mathcal{J}_\tau)$ for all $\tau \in [d]$ to be performed in $\mathcal{O}(1)$ time, once $\mathbb{E}\Gamma$ and $\sum_{j \in \mathcal{J}_\tau} p_j$ are precomputed. In a sharp contrast, evaluating $\pi(\mathcal{J}_\tau)$ for any $\tau \in [d]$ requires $\mathcal{O}(d)$ operations each time. Notably, the first result, (3) in Theorem 2, credited to Dai and Li [2023] and built upon more fundamental results of reciprocal moments Chao and Strawderman [1972], Wooff [1985], is quite general and may be of independent interest for other applications.

The approximation error bound (5), particularly when $\mathcal{I} = \mathcal{J}_\tau$, decreases as the expected volume of predicted mask increases, which typically occurs when $d$ is large. Even for small objects occupying only a $30 \times 30$ region in a $256 \times 256$ image, with an expected volume $\mathbb{E}(\Gamma) = \tau = 1000$, the approximation error remains below $0.1\%$, which is generally acceptable in practice.

We now summarize RankDice-RMA for binary segmentation in Algorithm 1. RankIoU-RMA is developed analogously in Section B, with the same approximation. The first two steps prepare and store intermediate values for evaluating $\widehat{\pi}_{\text{RMA}}(\widehat{\mathcal{J}}_\tau)$ based on an estimated probabilities $\widehat{\mathbf{p}}$. After that, we identify the optimal volume $\widehat{\tau}^*$ and make prediction by selecting the top $\widehat{\tau}^*$ pixels. Neglecting the sorting operation, the time complexity of the RankDice-RMA and RankIoU-RMA is reduced to $\mathcal{O}(d)$, compared to $\mathcal{O}(d \log d)$ for RankDice-BA and $\mathcal{O}(d^2)$ for RankIoU. For example, RankDice-RMA achieves 48x speedup for RankDice-BA in LiTS dataset [Bilic et al., 2023].

---

**Algorithm 1** RankDice-RMA-Binary

---

**Input:** Estimated probability map $\widehat{\mathbf{p}} \in [0,1]^d$ for a given input image.

**Output:** The predicted segmentation mask $\widehat{\boldsymbol{\delta}} \in \{0,1\}^d$.

 1: Rank probabilities $\widehat{\mathbf{p}}$ in descending order, yielding $\widehat{p}_{j_1} \geq \cdots \geq \widehat{p}_{j_d}$.
 2: Prepare cumulative sum of top probabilities and mean of Poisson-binomial

$$\widehat{q}_\tau = \sum_{k=1}^{\tau} \widehat{p}_{j_k} \quad \text{for } \tau \in [d], \quad \widehat{\mu} = \sum_{j=1}^{d} \widehat{p}_j.$$

 3: Compute $\widehat{\pi}_{\text{RMA}}(\widehat{\mathcal{J}}_\tau) = \frac{2\widehat{q}_\tau}{\tau + \widehat{\mu} + 1}$ for $\tau \in [d]$, according to (4).
 4: Determine optimal volume $\widehat{\tau}^* = \operatorname{argmax}_{\tau \in [d]} \widehat{\pi}_{\text{RMA}}(\widehat{\mathcal{J}}_\tau)$.
 5: Make prediction by $\widehat{\delta}_j = \mathbb{1}(p_j \geq \widehat{p}_{j_{\widehat{\tau}^*}})$ for $j \in [d]$.

---

## 3.2 RMA-score for non-overlapping multiclass segmentation

To extend RankSEG to non-overlapping multiclass segmentation, a natural approach is first applying binary RankSEG to each class independently, and then address any overlaps. As discussed in the introduction, perfectly addressing overlaps is currently beyond the capabilities of RankSEG, as the non-overlapping constraint leads to a nonlinear assignment problem [Kuhn, 1955], which is generally computationally intractable. Therefore, the focus of this section is on utilizing RMA to solve overlapping pixels provided by RankSEG, ultimately producing non-overlapping segmentation.

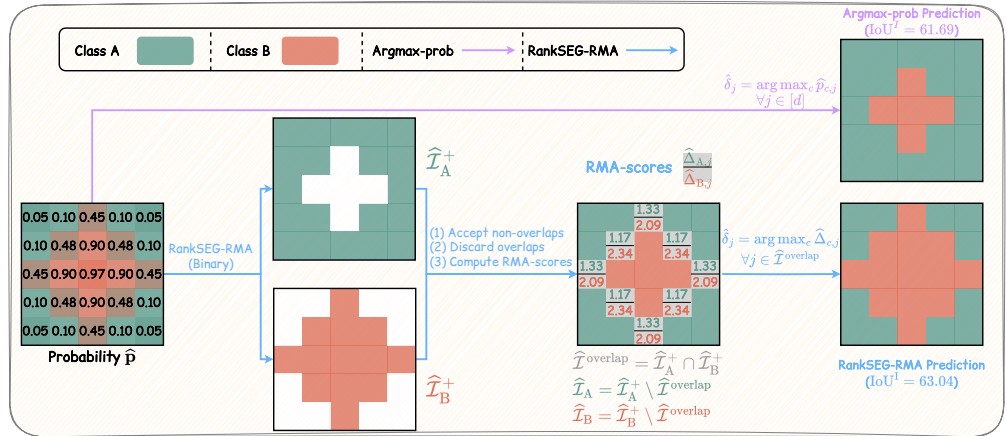

Figure 1: Comparison of Argmax-prob and RankSEG-RMA. (a) Argmax-prob: each pixel is predicted to the class with the highest probability. (b) RankSEG-RMA: segmentation masks $\widehat{\mathcal{I}}_c^+$ for each class are obtained independently; non-overlapping parts $\widehat{\mathcal{I}}_c$ are accepted, while overlaps $\widehat{\mathcal{I}}^{\text{overlap}}$ are resolved by applying argmax over RMA-scores.

Our solution draws inspiration from *Argmax-prob* method, which efficiently resolves non-overlapping constraint by assigning pixels to their highest-probability classes, that is,

$$\widehat{\delta}_j = \operatorname*{argmax}_{c \in [C]} \widehat{p}_{c,j} \quad \forall j \in [d],$$

where $\widehat{p}_{c,j}$ is the estimated probability of pixel $j$ belonging to class $c$. This approach is computationally efficient but does not guarantee optimal performance. One reason is that merely examining probability values does not accurately reflect how individual pixel assignments contribute to segmentation metrics. To address this issue, we extend the probability in the argmax framework to a Dice/IoU-related score. Unlike the probability score, our proposed score function is grounded on the Bayes rule in Theorem 1 and can be efficiently computed by RMA. We refer to the score function as *RMA-score*.

---

**Algorithm 2** RankDice-RMA-Multiclass

---

**Input:** Estimated probability map $\widehat{\mathbf{p}} \in [0,1]^{C \times d}$.

**Output:** The predicted segmentation mask $\widehat{\boldsymbol{\delta}} \in [C]^d$.

1: /* Obtain overlapping segmentation mask */
2: **for** $c = 1$ to $C$ **do**
3:    $\widehat{\psi}_c = \texttt{RankDice-RMA-Binary}(\widehat{\mathbf{p}}_c), \widehat{\mathcal{I}}_c^+ = \{j : \widehat{\psi}_{c,j} = 1\}$.
4: **end for**
5: /* Resolve overlapping by argmax over RMA-scores */
6: Identify overlapping indices, $\widehat{\mathcal{I}}^{\text{overlap}} = \bigcup_{c \neq c'} (\widehat{\mathcal{I}}_c^+ \cap \widehat{\mathcal{I}}_{c'}^+)$.
7: **for** $c = 1$ to $C$ **do**
8:    Discard assignments for overlapping pixels, $\widehat{\mathcal{I}}_c = \widehat{\mathcal{I}}_c^+ \setminus \widehat{\mathcal{I}}^{\text{overlap}}$.
9:    Accept prediction for not overlapping pixels, $\widehat{\delta}_j = c$ for $j \in \widehat{\mathcal{I}}_c$.
10: **end for**
11: Compute RMA-scores, $\widehat{\Delta}_{c,j}$ via (7) for $j \in \widehat{\mathcal{I}}^{\text{overlap}}$ and $c \in [C]$.
12: Resolve overlapping by argmax, $\widehat{\delta}_j = \text{argmax}_{c \in [C]} \widehat{\Delta}_{c,j}$ for $j \in \widehat{\mathcal{I}}^{\text{overlap}}$.
13: **Return** $\widehat{\boldsymbol{\delta}}$

---

To proceed, let $\widehat{\mathcal{I}}_c^+$ denote the index set of pixels assigned to class $c$ by RankSEG, $\widehat{\mathcal{I}}^{\text{overlap}} = \bigcup_{c \neq c'} (\widehat{\mathcal{I}}_c^+ \cap \widehat{\mathcal{I}}_{c'}^+)$ the index set of overlapping pixels, and $\widehat{\mathcal{I}}_c = \widehat{\mathcal{I}}_c^+ \setminus \widehat{\mathcal{I}}^{\text{overlap}}$ the non-overlapping part, as illustrated in Figure 1. We resolve $\widehat{\mathcal{I}}^{\text{overlap}}$ to ensure segmentation masks are non-overlapping:

$$\widehat{\delta}_j = \underset{c \in [C]}{\text{argmax}} \, \widehat{\Delta}_{c,j}, \quad \forall j \in \widehat{\mathcal{I}}^{\text{overlap}}, \tag{6}$$

where $\widehat{\Delta}_{c,j}$ is increment of Dice-RMA by adding pixel $j$ for class $c$, which is defined as:

$$\widehat{\Delta}_{c,j} = \widehat{\pi}_{\text{RMA}}(\widehat{\mathcal{I}}_c \cup \{j\}) - \widehat{\pi}_{\text{RMA}}(\widehat{\mathcal{I}}_c) = \frac{2 \left( \widehat{p}_{c,j} + \sum_{i \in \widehat{\mathcal{I}}_c} \widehat{p}_{c,i} \right)}{|\widehat{\mathcal{I}}_c| + \widehat{\mu}_c + 2} - \frac{2 \sum_{i \in \widehat{\mathcal{I}}_c} \widehat{p}_{c,i}}{|\widehat{\mathcal{I}}_c| + \widehat{\mu}_c + 1}, \tag{7}$$

where $\widehat{\mu}_c = \sum_{j=1}^d \widehat{p}_{c,j}$ represents the estimated mean volume of class $c$. Intuitively, (6) maximizes the immediate improvement by choosing the class that yields the highest marginal gain in the Dice-RMA objective. While this greedy solution does not guarantee a globally optimal assignment over all overlapping pixels simultaneously, it is computationally efficient and empirically effective for reducing overlap and improving final segmentation performance.

To summarize, the procedure of RankDice-RMA for multiclass segmentation is presented in Algorithm 2. After applying binary RankDice-RMA, the predicted set $\widehat{\mathcal{I}}_c^+$ of each class $c$ is obtained. The overlapping pixels $\widehat{\mathcal{I}}^{\text{overlap}}$ are then identified, and the assignments for these pixels are discarded. The non-overlapping pixels $\widehat{\mathcal{I}}_c$ are assigned to their respective classes. Finally, we compute the RMA-scores for the overlapping pixels and resolve overlaps by selecting the class with the highest score. The complexity for addressing overlapping is $\mathcal{O}(Cd)$, which is no worse than Argmax-prob.

## 4 Experiments

### 4.1 Setup

**Datasets.** We conduct experiments on five datasets: (1) PASCAL VOC [Everingham et al., 2010], (2) Cityscapes [Cordts et al., 2016], (3) ADE20K [Zhou et al., 2017], (4) LiTS [Bilic et al., 2023], and (5) KiTS [Heller et al., 2021]. These datasets cover a diverse range of scenarios, including urban scenes (Cityscapes), "thing" and "stuff" (PASCAL VOC and ADE20K), as well as medical images (LiTS and KiTS). The datasets contain between 200 images (LiTS) and 20,000 images (ADE20K), and the number of classes varies from binary segmentation (LiTS) to over a hundred (ADE20K). We only segment tumors in LiTS and KiTS, treating them as binary segmentation tasks.

**Models.** We employ following six segmentation models: (1) UNet [Ronneberger et al., 2015], (2) DeepLabV3+ [Chen et al., 2018], (3) PSPNet [Zhao et al., 2017], (4) UPerNet [Xiao et al.,

2018], (5) SegFormer [Xie et al., 2021], and (6) CPT [Tang et al., 2025]. The first four models are CNN-based and utilize backbones such as ResNet [He et al., 2016] or ConvNeXt [Liu et al., 2022], whereas SegFormer and CPT are transformer-based models. The models are trained using the cross-entropy loss, and we compare the proposed RankSEG-RMA with the conventional argmax or thresholding rule for multiclass or binary segmentation, respectively. The training details can be found in Section D.

**Evaluation.** As discussed in Section 2.1, we evaluate the segmentation models using both $mIoU^I/mDice^I$ and $mIoU^C/mDice^C$, which are straightforward extensions of binary metric $IoU^I/Dice^I$ to multiclass segmentation. The metrics with superscripts $^I$ and $^C$ differ when not all classes are present in every image (see Wang et al. [2023a] for details).

## 4.2 Overall performance

Table 1: Performance for different prediction methods with various models on PASCAL VOC, Cityscapes, and ADE20K.

| Model | Prediction | PASCAL VOC | | | | Cityscapes | | | | ADE20K | | | |
|---|---|---|---|---|---|---|---|---|---|---|---|---|---|
| | | $mIoU^I$ | $mIoU^C$ | $mDice^I$ | $mDice^C$ | $mIoU^I$ | $mIoU^C$ | $mDice^I$ | $mDice^C$ | $mIoU^I$ | $mIoU^C$ | $mDice^I$ | $mDice^C$ |
| PSPNet | Argmax-prob | 83.59 | 72.59 | 87.69 | 78.22 | 71.33 | 63.38 | 78.96 | 71.34 | 49.78 | 33.83 | 56.89 | 40.36 |
| (ResNet50) | RankDice-RMA | 84.21 | 73.91 | 88.42 | 79.75 | 72.00 | 64.20 | 79.68 | 72.28 | 50.70 | 36.30 | 58.52 | 43.67 |
| PSPNet | Argmax-prob | 85.48 | 75.57 | 89.18 | 80.78 | 73.07 | 65.89 | 80.45 | 73.55 | 51.32 | 37.42 | 58.66 | 44.44 |
| (ResNet101) | RankDice-RMA | 85.98 | 76.64 | 89.74 | 81.94 | 73.72 | 66.53 | 81.14 | 74.28 | 51.57 | 38.09 | 59.17 | 45.29 |
| DeepLabV3+ | Argmax-prob | 84.19 | 73.96 | 88.11 | 79.31 | 73.55 | 65.98 | 80.80 | 73.63 | 49.78 | 33.83 | 56.89 | 40.36 |
| (ResNet50) | RankDice-RMA | 84.79 | 75.26 | 88.84 | 80.88 | 74.05 | 66.68 | 81.38 | 74.49 | 49.82 | 34.28 | 57.19 | 40.92 |
| DeepLabV3+ | Argmax-prob | 86.40 | 77.25 | 89.83 | 82.08 | 73.37 | 66.17 | 80.59 | 73.71 | 52.53 | 37.52 | 59.57 | 44.13 |
| (ResNet101) | RankDice-RMA | 86.80 | 78.14 | 90.32 | 83.14 | 73.92 | 66.68 | 81.24 | 74.33 | 52.64 | 38.14 | 59.95 | 44.85 |
| SegFormer | Argmax-prob | 85.40 | 75.85 | 89.21 | 81.13 | 73.24 | 65.57 | 80.49 | 73.16 | 53.03 | 38.19 | 60.06 | 44.83 |
| (MiTB2) | RankDice-RMA | 85.85 | 76.01 | 89.44 | 81.04 | 73.81 | 66.41 | 81.14 | 74.13 | 53.67 | 39.09 | 61.09 | 46.11 |
| SegFormer | Argmax-prob | 86.86 | 77.57 | 90.11 | 82.15 | 73.32 | 66.13 | 80.53 | 73.65 | 54.09 | 40.00 | 61.03 | 46.50 |
| (MiTB4) | RankDice-RMA | 87.28 | 78.59 | 90.56 | 83.22 | 74.10 | 67.14 | 81.38 | 74.74 | 54.72 | 40.82 | 61.92 | 47.57 |
| UPerNet | Argmax-prob | 87.82 | 79.52 | 91.03 | 84.11 | 75.66 | 68.83 | 82.61 | 76.08 | 56.94 | 42.86 | 63.98 | 49.61 |
| (ConvNeXt) | RankDice-RMA | 88.25 | 80.31 | 91.48 | 84.98 | 76.17 | 69.57 | 83.21 | 76.97 | 57.67 | 43.84 | 64.93 | 50.85 |
| CPT | Argmax-prob | 88.56 | 80.74 | 91.62 | 85.18 | 75.33 | 68.39 | 82.25 | 75.74 | 57.85 | 44.59 | 64.75 | 51.27 |
| (Swin-Large) | RankDice-RMA | 88.89 | 81.53 | 92.01 | 86.08 | 75.86 | 69.29 | 82.85 | 76.76 | 58.63 | 45.56 | 65.83 | 52.58 |

Table 2: Performance for different prediction methods with various models on LiTS and KiTS.

| Prediction | Model | LiTS | | KiTS | | Model | LiTS | | KiTS | |
|---|---|---|---|---|---|---|---|---|---|---|
| | | $IoU^I$ | $Dice^I$ | $IoU^I$ | $Dice^I$ | | $IoU^I$ | $Dice^I$ | $IoU^I$ | $Dice^I$ |
| Argmax-prob | DeepLabV3+ | 34.31 | 42.81 | 54.61 | 47.20 | UNet | 36.40 | 45.18 | 56.03 | 49.28 |
| RankDice-BA | (ResNet18) | 36.11 | 45.04 | 58.00 | 50.57 | (ResNet18) | 38.34 | 47.54 | 59.10 | 52.08 |
| RankDice-RMA | | 36.12 | 45.04 | 58.00 | 50.57 | | 38.34 | 47.54 | 59.10 | 52.08 |
| Argmax-prob | DeepLabV3+ | 38.45 | 47.58 | 61.16 | 54.19 | UNet | 38.45 | 47.58 | 57.36 | 51.00 |
| RankDice-BA | (ResNet50) | 40.09 | 49.50 | 63.56 | 56.22 | (ResNet50) | 40.71 | 50.08 | 60.07 | 50.34 |
| RankDice-RMA | | 40.09 | 49.50 | 63.56 | 56.22 | | 40.70 | 50.07 | 60.07 | 53.54 |

Table 3: Time consumption (in seconds) of model forward and different prediction rules with single A100 GPU. DeepLabV3+ (ResNet50) is used for the medical datasets, while UPerNet (ConvNeXt) is used for the others. The mean and standard deviation over 10 runs are reported. ✗ indicates that the method is not applicable due to non-overlapping benchmark setups.

| | Pascal VOC | Cityscapes | ADE20K | LiTS | KiTS |
|---|---|---|---|---|---|
| Argmax-prob | 0.05 ($\pm$0.01) | 0.22 ($\pm$0.01) | 0.43 ($\pm$0.08) | 0.01 ($\pm$0.00) | 0.01 ($\pm$0.00) |
| RankDice-RMA | 6.93 ($\pm$1.14) | 10.15 ($\pm$1.77) | 58.00 ($\pm$3.44) | 0.34 ($\pm$1.19) | 0.26 ($\pm$0.15) |
| RankDice-BA | ✗ | ✗ | ✗ | 16.33 ($\pm$1.19) | 9.99 ($\pm$0.15) |
| Model forward | 40.77 ($\pm$5.15) | 175.81 ($\pm$3.02) | 324.59 ($\pm$13.42) | 14.15 ($\pm$0.64) | 11.86 ($\pm$0.20) |

Results for PASCAL VOC, Cityscapes, and ADE20k are presented in Table 1, while those for LiTS and KiTS are shown in Table 2. The best performance within each model is highlighted in **bold**, while the best across all models is highlighted in pink. If two performances are very close and both are the best, we highlight both. Three observations can be drawn from these results.

- **Our proposed method significantly outperforms the conventional Argmax-prob across all datasets and models**, irrespective of light or heavy backbones, demonstrating its effectiveness and robustness. For instance, on Cityscapes with SegFormer (MiTB4), RankDice-RMA improves mDice$^I$ and mDice$^C$ by 0.85% and 1.09%, respectively. In addition, on LiTS with UNet (ResNet50), RankDice-RMA outperforms Argmax-prob by 2.49% in Dice$^I$.

- As shown in Tables 2 and 3, RankDice-RMA achieves significant time efficiency improvements over RankDice-BA while maintaining similar performance on the LiTS (48x speedup) and KiTS datasets (38x speedup). Hence, we conclude that **RankDice-RMA is a strict improvement over RankDice-BA**. Although RankDice-RMA is slower than Argmax-prob, the absolute time consumption is negligible compared to the model forward time. In contrast, such argument can not be applied to RankDice-BA, whose time consumption is comparable to the model forward.

- **RankDice-RMA simultaneously boost IoU performance**, even though it is originally motivated by the Bayes rule for Dice. Furthermore, RankDice-RMA and RankIoU-RMA achieve nearly identical performance across all experiments (results for RankIoU-RMA are omitted for simplicity), suggesting that the two metrics are closely related and that either RankDice-RMA or RankIoU-RMA can serve as a unified prediction method for both metrics.

## 4.3 Class-wise performance

To further evaluate the performance of RankDice-RMA, we report class-wise results on PASCAL VOC in Table 4. Improvements over Argmax-prob are highlighted in **green** for positive changes and in **red** for negative ones. The results indicate that RankDice-RMA consistently enhances performance across most classes. Two key observations can be made:

- The performance gains are more pronounced for classes with lower baseline performance, suggesting that **RankDice-RMA is particularly effective for difficult classes**. For instance, the `Chair` class, which exhibits a low IoU of 48.05% under Argmax-prob, is boosted to 50.52%, an enhancement of 2.47%; whereas the `Areoplane` class, with a high initial IoU of 90.39%, only sees a marginal improvement of 0.45%. This trend may result in negative changes for classes like `Bird` and `Sheep`, where Argmax-prob already performs well, leaving limited room for improvement with the Bayes rule.

- Although the error bound in Theorem 2 implies a larger approximation error for classes with smaller volume, the results show that **RankDice-RMA still achieves substantial performance gains for these small objects**. For example, our analysis indicates that `Bottle` and `Chair` are among the smallest objects in the dataset. Nonetheless, these classes exhibit significant improvements, possibly because the benefits of the Bayes rule outweigh the approximation error.

Table 4: Class-wise IoU on PASCAL VOC with UPerNet (ConvNeXt).

| Prediction | Aeroplane | Bicycle | Bird | Boat | Bottle | Bus | Car | Cat | Chair | Cow |
|---|---|---|---|---|---|---|---|---|---|---|
| Argmax-prob | 90.39 | 50.33 | 91.18 | 81.19 | 69.21 | 89.55 | 78.78 | 92.24 | 48.05 | 92.47 |
| RankDice-RMA | 90.84 | 51.76 | 90.86 | 81.70 | 71.77 | 90.31 | 80.46 | 92.43 | 50.52 | 92.69 |
| *(Improvement)* | **+0.45** | **+1.43** | **-0.32** | **+0.51** | **+2.56** | **+0.76** | **+1.68** | **+0.19** | **+2.47** | **+0.22** |

| Prediction | Dining Table | Dog | Horse | Motorbike | Person | Potted Plant | Sheep | Sofa | Train | TV Monitor |
|---|---|---|---|---|---|---|---|---|---|---|
| Argmax-prob | 54.12 | 93.55 | 91.21 | 89.51 | 82.26 | 57.11 | 92.59 | 62.09 | 91.97 | 77.25 |
| RankDice-RMA | 55.28 | 93.56 | 91.34 | 89.80 | 83.55 | 59.20 | 92.54 | 62.88 | 92.17 | 78.05 |
| *(Improvement)* | **+1.16** | **+0.01** | **+0.13** | **+0.29** | **+1.29** | **+2.09** | **-0.05** | **+0.79** | **+0.20** | **+0.80** |

## 4.4 Worst-case analysis

For safe-critical applications, it is crucial to evaluate the worst-case performance of segmentation models. In this context, image-level metrics provide more detailed insights than dataset-level metrics for assessing worst-case scenarios [Wang et al., 2023a]. Without loss of generality, consider $\text{mIoU}_1^I \leq \text{mIoU}_2^I \leq \cdots \leq \text{mIoU}_n^I$ denote the sorted image-level mIoU values for $n$ images in a test set. We define the average mIoU over those below the lowest $q$-th quantile as:

$$\text{mIoU}^{I_q} = \frac{1}{\lfloor nq \rfloor} \sum_{i=1}^{\lfloor nq \rfloor} \text{mIoU}_i^I.$$

By definition, this metric quantifies performance of the worst $q$-th quantile images. Table 5 presents the mIoU$^{I_5}$ and mIoU$^{I_{10}}$ results, where UPerNet(ConvNeXt) is used for PASCAL VOC, Cityscapes, and ADE20K, while DeepLabV3+(ResNet50) is used for LiTS and KiTS. The results demonstrate that **our method also improves the worst-case performance across all datasets**.

Table 5: mIoU$^{I_5}$ and mIoU$^{I_{10}}$ on PASCAL VOC, Cityscapes, ADE20K, LiTS, and KiTS.

| Prediction | mIoU$^{I_5}$ | | | | | mIoU$^{I_{10}}$ | | | | |
|---|---|---|---|---|---|---|---|---|---|---|
| | VOC | Cityscapes | ADE20K | LiTS | KiTS | VOC | Cityscapes | ADE20K | LiTS | KiTS |
| Argmax-prob | 44.80 | 59.02 | 24.99 | 2.13 | 6.72 | 52.10 | 61.72 | 29.54 | 4.05 | 8.39 |
| RankDice-RMA | **46.21** | **59.96** | **25.56** | **2.70** | **8.49** | **53.17** | **62.51** | **30.35** | **4.81** | **10.38** |
| *(Improvement)* | **+1.41** | **+0.94** | **+0.57** | **+0.57** | **+1.77** | **+1.07** | **+0.79** | **+0.81** | **+0.76** | **+1.89** |

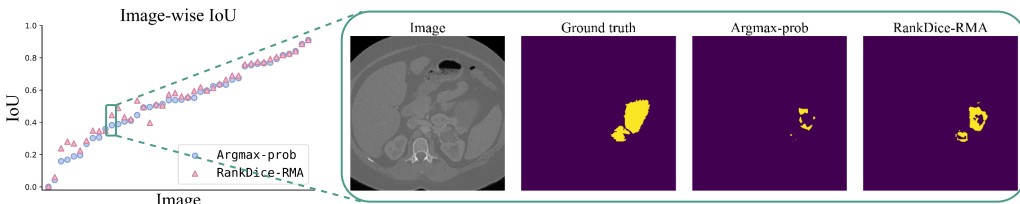

Figure 2: Image-wise performance and an example of a worst-case segmentation on KiTS. The left plot presents the IoU for each image, with indices sorted in ascending order according to IoU under Argmax-prob. The right plot displays the segmentation results for a slice of a worst-case image.

Figure 2 shows image-level IoU for a KiTS validation fold and a worst-case segmentation example. The left plot shows IoU values for each image. It is evident that **our method outperforms Argmax-prob across most images, especially on difficult cases**. The right plot displays segmentation results for one worst-case image. The tumor consists of two adjacent segments, but Argmax-prob captures only a small portion of the larger segment and almost misses the smaller one. In contrast, **our method not only produces a more complete segmentation but also successfully identifies the smaller segment**. This example highlights the potential of our method for challenging clinical scenarios.

## 4.5 Ablation studies

**Effect of the RMA-score.** We have introduced the RMA-score to address $\widehat{\mathcal{I}}^{\text{overlap}}$ that occurs when applying RankDice for each class independently. We now demonstrate that the scores are indeed crucial for improving performance by comparing them with two ad-hoc alternatives in (6):

- **Prob-scores.** The predicted probability $\widehat{p}_{c,j}$ is directly used as score of pixel $j$ for class $c$.
- **WProb-scores.** As inspired by the RMA-scores or intuitive reasoning, classes with more already predicted pixels should have lower preference when resolving overlaps. Hence, a weighted version of the predicted probabilities is considered, i.e., $\widehat{s}_{c,j} = \widehat{p}_{c,j}/|\widehat{\mathcal{I}}_c|$.

As shown in Table 6, WProb-scores outperform Prob-scores on Pascal VOC and Cityscapes, supporting the intuition to account for predicted volume. However, WProb-scores underperform on ADE20K, indicating that simple weighting fails when many classes are present. In contrast,

Table 6: mIoU$^I$ of using different scores.

| | Pascal VOC | Cityscapes | ADE20K |
|---|---|---|---|
| Prob-scores | 87.83 | 75.75 | 56.96 |
| WProb-scores | 88.17 | 75.89 | 56.75 |
| RMA-scores | **88.25** | **76.17** | **57.67** |

RMA-scores consistently perform best, particularly on ADE20K, where overlapping phenomena are more complex due to the large number of classes. This superiority is due to that RMA-scores are derived from the Bayes rule, making them more principled than heuristic methods. These results support our claim that RMA-scores are essential for improved performance.

**Effect of different bounds in RMA.** Recall that Theorem 2 provides both lower and upper bounds under RMA, with the lower bound being preferred for its simplicity. As a complement, we further find that using the upper bound as an alternative approximation yields same performance. This suggests that **the choice of different bounds does not bother**. More importantly, this confirms that the bounds are tight, aligning with our theoretical analysis in Theorem 2.

# 5 Conclusion

In this paper, we propose RankSEG-RMA, a novel segmentation algorithm that grounds on the Bayes rule, and enjoys computational efficiency by using reciprocal moment approximation (RMA). Extensive experiments across various datasets and models demonstrate that RankSEG-RMA outperforms the conventional Argmax-prob and significantly reduces computational cost compared to the existing RankSEG-BA. Nevertheless, two limitations are noteworthy for future improvement. First, the proposed overlap resolution method predicts each pixel independently, which may not be optimal; future work could explore more global approaches while maintaining computational efficiency. Second, our work builds upon the assumption of conditional independence in Bayes rule, which could be relaxed in subsequent research.

## Acknowledgments

We thank the anonymous Area Chair and reviewers for their valuable feedback, suggestions and support. This work was supported by the Hong Kong RGC-ECS Grant 24302422 and Hong Kong RGC Grant 14304823.

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

# Table of Contents

## A  Segmentation calibration and RankSEG Framework

Given a training dataset $\{(\mathbf{x}_i, \mathbf{y}_i)\}_{i=1}^n$ where $\mathbf{x}_i \in \mathcal{X}$, $\mathbf{y}_i \in \mathcal{Y}$, a loss function $\ell : \mathcal{Y} \times \mathcal{Y} \to \mathbb{R}$, and a hypothesis class $\mathcal{H} = \{h : \mathcal{X} \to \mathcal{Y}\}$, the empirical risk and population risk are defined as:

$$\widehat{\mathcal{R}}_\ell(h) = \frac{1}{n} \sum_{i=1}^n \ell(h(\mathbf{x}_i), \mathbf{y}_i) \quad \text{and} \quad \mathcal{R}_\ell(h) = \mathbb{E}_{\mathbf{X},\mathbf{Y}}[\ell(h(\mathbf{X}), \mathbf{Y})].$$

The empirical risk minimizer $\widehat{h}_n = \operatorname{argmin}_{h \in \mathcal{H}} \widehat{\mathcal{R}}_\ell(h)$ is used for making predictions. However, it is often the case that the target loss function is neither differentiable and nor convex, such as the zero-one loss in classification or the negative of IoU$^{\mathrm{I}}$/Dice$^{\mathrm{I}}$ in our case, making direct optimization infeasible. Therefore, a surrogate loss function $\phi : \mathcal{Z} \times \mathcal{Y} \to \mathbb{R}$, combined with a surrogate hypothesis class $\mathcal{F} = \{f : \mathcal{X} \to \mathcal{Z}\}$ and a decoding function (also known as link function) $d : \mathcal{Z} \to \mathcal{Y}$, is typically employed:

$$\widehat{f}_n = \operatorname*{argmin}_{f \in \mathcal{F}} \widehat{\mathcal{R}}_\phi \quad \text{and} \quad \bar{h}_n = d \circ \widehat{f}_n.$$

Note that the surrogate loss is designed to be easier to optimize than the original loss, the output space of the surrogate hypothesis may not align with the label space, and the decoding function maps the surrogate prediction back to the original label space. The desired property of the surrogate loss is calibrated, as specified in Theorem A.1.

**Definition A.1** (Calibration). A surrogate loss $\phi$, associated with a decoding function $d$, is calibrated with respect to a target loss $\ell$ if, for any distribution over $\mathcal{X} \times \mathcal{Y}$ and any sequences $\{f_n\}_{n \in \mathbb{N}} \subset \mathcal{F}$, the following holds:

$$\left( \mathcal{R}_\phi(\widehat{f}_n) \to \inf_{f \in \mathcal{F}} \mathcal{R}_\phi(f) \right) \quad \implies \quad \left( \mathcal{R}_\ell(d \circ \widehat{f}_n) \to \inf_{h \in \mathcal{H}} \mathcal{R}_\ell(h) \right) \quad \text{as} \quad n \to \infty.$$

For example, hinge loss with sign as decoding function and cross entropy loss with argmax as decoding function are calibrated with respect to zero-one loss in binary classification and multiclass classification, respectively [Lin, 2004, Zhang, 2004, Bartlett et al., 2006, Tewari and Bartlett, 2007, Mao et al., 2023].

In general, there are two principled approaches to achieving calibration or consistency: (1) designing a consistent surrogate loss function and making predictions via a suitable decoding function [Bartlett et al., 2006, Tewari and Bartlett, 2007], and (2) directly deriving the Bayes rule for target metrics and plugging in the estimated probabilities for prediction [Nowozin, 2014, Dembczynski et al., 2013, Dai and Li, 2023].

RankSEG [Dai and Li, 2023] belongs to the latter category. It does not require a carefully designed surrogate loss function and can be directly applied to models trained with cross-entropy loss; however, the decoding step is more involved. Nevertheless, Dai and Li [2023] demonstrate a ranking property of the Bayes rule, stating that the optimal prediction is to select the top $\tau^*$ pixels with the highest conditional probabilities, which significantly simplifies the decoding step.

# B    RankIoU-RMA

**Theorem B.1** (The Bayes rule for IoU$^{\text{I}}$-segmentation [Dai and Li, 2023]). *Assume that $Y_i \perp Y_j | \mathbf{X}$. A segmentation rule $\boldsymbol{\delta}^*$ is a global maximizer of $\mathbb{E}(IoU^I(\boldsymbol{\delta}))$ if and only if $\delta_j^* = \mathbb{1}(p_j \geq p_{j_{\tau^*}})$, and $j_\tau$ is the index with $\tau$-th largest probability. The optimal volume $\tau^*$ is given by:*

$$\tau^* = \underset{\tau \in \{0,1,\cdots,d\}}{\operatorname{argmax}} \nu(\mathcal{J}_\tau) \quad with \quad \nu(\mathcal{J}_\tau) = \left( \sum_{j \in \mathcal{J}_\tau} p_j \right) \mathbb{E}\left( \frac{1}{\tau + \Gamma_{-\mathcal{J}_\tau}} \right) \tag{8}$$

*where $\mathcal{J}_\tau = \left\{ j' : p_{j'} \geq p_{j_\tau} \right\}$ is the index set of the top $\tau$ conditional probabilities with $\mathcal{J}_0 = \emptyset$, and $\Gamma_{-\mathcal{J}_\tau} = \sum_{j' \notin \mathcal{J}_\tau} B_{j'}$ is Poisson-binomial random variable with $B_{j'}$ being a Bernoulli random variable with success probability $p_{j'}$.*

According to Theorem B.1, the Bayes rule for IoU$^{\text{I}}$ shares substantial similarity with that of Dice$^{\text{I}}$, both of which consist of two parts: (1) ranking the conditional probabilities and (2) selecting the top $\tau^*$ pixels as positives. The primary difference lies in the computation of score functions when determining of the optimal volume $\tau^*$, which is tailored to the respective metric. The consistency of RankSEG [Dai and Li, 2023, Lemma 10] is established by plugging in the estimated probabilities $\widehat{p}_j(\mathbf{x}; \theta)$, where $\theta$ is the model parameter trained by minimizing a strictly proper loss [Gneiting and Raftery, 2007].

Note that replacing $\Gamma_{-\mathcal{J}_\tau}$ with $\Gamma$ in Theorem B.1 leads to large approximation error, especially when $\tau$ is large. Therefore, Blind approximation is no longer applicable in this case. However, RMA technique can still be employed to approximate $\nu(\mathcal{J}_\tau)$:

$$\nu_{\text{RMA}}(\mathcal{J}_\tau) = \left( \sum_{j \in \mathcal{J}_\tau} p_j \right) \frac{1}{\tau + \mathbb{E}(\Gamma_{-\mathcal{J}_\tau})} \tag{9}$$

Based on this, we develop RankIoU-RMA for binary segmentation, as described in Algorithm 3. This algorithm is highly similar to RankDice-RMA, with the only difference being the use of the target function $\widehat{\nu}(\widehat{\mathcal{J}}_\tau)$.

---

**Algorithm 3** RankIoU-RMA-Binary

---

**Input:** Estimated probability map $\widehat{\mathbf{p}} \in [0,1]^d$.
**Output:** The predicted segmentation mask $\widehat{\boldsymbol{\delta}} \in \{0,1\}^d$.
1:  Rank probabilities $\widehat{\mathbf{p}}$ in descending order, yielding $\widehat{p}_{j_1} \geq \cdots \geq \widehat{p}_{j_d}$.
2:  Prepare cumulative sum of top probabilities and mean of Poisson-binomial

$$\widehat{q}_\tau = \sum_{k=1}^{\tau} \widehat{p}_{j_k} \quad \text{for } \tau \in [d], \quad \widehat{\mu} = \sum_{j=1}^{d} \widehat{p}_j.$$

3:  Compute $\widehat{\nu}_{\text{RMA}}(\widehat{\mathcal{J}}_\tau) = \frac{\widehat{q}_\tau}{\tau + (\widehat{\mu} - \widehat{q}_\tau)}$ for $\tau \in [d]$, according to (9).
4:  Determine optimal volume $\widehat{\tau}^* = \operatorname{argmax}_{\tau \in [d]} \widehat{\nu}_{\text{RMA}}(\widehat{\mathcal{J}}_\tau)$.
5:  Make prediction by $\widehat{\delta}_j = \mathbb{1}(p_j \geq \widehat{p}_{j_{\widehat{\tau}^*}})$ for $j \in [d]$.

---

In order to extend RankIoU-RMA to non-overlapping multiclass segmentation, it suffices to use the following RMA-scores for IoU, followed by an argmax to resolve overlaps:

$$\widehat{\Omega}_{c,j} = \widehat{\nu}(\widehat{\mathcal{I}}_c \cup \{j\}) - \widehat{\nu}(\widehat{\mathcal{I}}_c) = \frac{\widehat{p}_{c,j} + \sum_{k \in \widehat{\mathcal{I}}_c} \widehat{p}_{c,k}}{|\widehat{\mathcal{I}}_c| + (\widehat{\mu}_c - \widehat{p}_{c,j} - \sum_{k \in \widehat{\mathcal{I}}_c} \widehat{p}_{c,k})} - \frac{\sum_{k \in \widehat{\mathcal{I}}_c} \widehat{p}_{c,k}}{|\widehat{\mathcal{I}}_c| + (\widehat{\mu}_c - \sum_{k \in \widehat{\mathcal{I}}_c} \widehat{p}_{c,k})}, \tag{10}$$

where $\widehat{\mathcal{I}}_c$ is the index set of pixels assigned to class $c$ and $\widehat{\mu}_c = \sum_{j=1}^{d} \widehat{p}_{c,j}$. The second term in (10) approximates the IoU when predicting mask by $\widehat{\mathcal{I}}_c$, while the first term approximates the IoU when pixel $j$ is further included. Similarly, Algorithm 4 can be obtained by simply replacing the RMA-scores used in RankDice-RMA-Multiclass.

---
**Algorithm 4** RankIoU-RMA-Multiclass
---
**Input:** Estimated probability map $\widehat{\mathbf{p}} \in [0,1]^{C \times d}$.
**Output:** The predicted segmentation mask $\widehat{\boldsymbol{\delta}} \in [C]^d$.

1: /* Obtain overlapping segmentation mask */
2: **for** $c = 1$ to $C$ **do**
3:    $\widehat{\psi}_c = \texttt{RankIoU-RMA-Binary}(\widehat{\mathbf{p}}_c)$, $\widehat{\mathcal{I}}_c^+ = \{j : \widehat{\psi}_{c,j} = 1\}$.
4: **end for**

5: /* Resolve overlapping by argmax over RMA-scores */
6: Identify overlapping indices, $\widehat{\mathcal{I}}^{\text{overlap}} = \bigcup_{c \neq c'} (\widehat{\mathcal{I}}_c^+ \cap \widehat{\mathcal{I}}_{c'}^+)$.
7: **for** $c = 1$ to $C$ **do**
8:    Discard assignments for overlapping pixels, $\widehat{\mathcal{I}}_c = \widehat{\mathcal{I}}_c^+ \setminus \widehat{\mathcal{I}}^{\text{overlap}}$.
9:    Accept prediction for not overlapping pixels, $\widehat{\delta}_j = c$ for $j \in \widehat{\mathcal{I}}_c$.
10: **end for**
11: Compute RMA-scores, $\widehat{\Omega}_{c,j}$ via (10) for $j \in \widehat{\mathcal{I}}^{\text{overlap}}$ and $c \in [C]$.
12: Resolve overlapping by argmax, $\widehat{\delta}_j = \text{argmax}_{c \in [C]} \widehat{\Omega}_{c,j}$ for $j \in \widehat{\mathcal{I}}^{\text{overlap}}$.

13: **Return** $\widehat{\boldsymbol{\delta}}$
---

## C    Proof of Theorem 2

The following two lemmas are used in the proof of (3) in Theorem 2.

**Lemma C.1** (Chao and Strawderman [1972])**.** *Let $a \in \mathbb{R}$ and $X$ be a random variable such that $X + a > 0$ a.s. Define the probability generating function of $X$ as $G_X(t) = \mathbb{E}(t^X)$ for $0 \leq t \leq 1$. Then,*

$$\mathbb{E}\left(\frac{1}{X+a}\right) = \int_0^1 G_X(u) t^{a-1} dt.$$

**Lemma C.2** (Wooff [1985])**.** *Let $\Lambda \sim \text{Bin}(d,p)$ be a binomial random variable. Then, for any $a > 0$, the following inequalities hold:*

$$\mathbb{E}\left(\frac{1}{\Lambda+a}\right) \leq \frac{1}{(d+1)p+a-1}.$$

Note that binomial random variable $\Lambda \sim \text{Bin}(d,p)$ and Poisson-binomial random variable $\Gamma \sim \text{PB}(p_1, p_2, \cdots, p_d)$ have probability generating functions:

$$G_\Lambda(t) = (1 - p + pt)^d \quad \text{and} \quad G_\Gamma(t) = \prod_{j=1}^d (1 - p_j + p_j t).$$

Now we are ready to prove Theorem 2.

*Proof.* We first prove (3). The lower bound that $(\mathbb{E}\Gamma + \tau)^{-1} \leq \mathbb{E}(\Gamma + \tau)^{-1}$ follows from the Jensen's inequality. Let $\Lambda \sim \text{Bin}(d, \bar{p})$, where $\bar{p} = d^{-1} \sum_{j=1}^d p_j$. To prove the upper bound, we have:

$$\mathbb{E}(\frac{1}{\Gamma + \tau}) = \int_0^1 t^{\tau-1} G_\Gamma(t) dt = \int_0^1 t^{\tau-1} \left( \prod_{j=1}^d (1 - p_j + p_j t) \right) dt$$

$$\leq \int_0^1 t^{\tau-1} (1 - \bar{p} + \bar{p}t)^d dt = \int_0^1 t^{\tau-1} G_\Lambda(t) dt = \mathbb{E}(\frac{1}{\Lambda + \tau})$$

$$\leq \left( \frac{1}{(d+1)\bar{p} + \tau - 1} \right).$$

The first and last equalities follow from Theorem C.1. The first inequality is due to the arithmetic and geometric means inequality, and the last inequality follows from Theorem C.2.

To proceed with (5), we first establish an error bound for RMA. Let $\Gamma$ be a Poisson-binomial random variable and let $\gamma \geq 1$. Then, we have:

$$\mathbb{E}(\Gamma + \tau)^{-1} - (\mathbb{E}\Gamma + \tau)^{-1} \leq (\frac{d+1}{d}\mathbb{E}\Gamma + \tau - 1)^{-1} - (\mathbb{E}\Gamma + \tau)^{-1}$$

$$\leq (\mathbb{E}\Gamma + \tau - 1)^{-1} - (\mathbb{E}\Gamma + \tau)^{-1} = \frac{1}{(\mathbb{E}\Gamma + \tau - 1)(\mathbb{E}\Gamma + \tau)} \leq (\mathbb{E}\Gamma + \tau)^{-2}. \quad (11)$$

For any $\mathcal{I} \subseteq [d]$, the error bound of RankDice-RMA is then given by:

$$|\pi_{\text{RMA}}(\mathcal{I}) - \pi(\mathcal{I})| \leq \sum_{j \in \mathcal{I}} 2p_j \left| \mathbb{E}(\frac{1}{\tau + \Gamma_{-j} + 1}) - \frac{1}{\tau + \mathbb{E}\Gamma + 1} \right|$$

$$= \sum_{j \in \mathcal{I}} 2p_j \left| \mathbb{E}(\frac{1}{\tau + \Gamma_{-j} + 1}) - \frac{1}{\tau + \mathbb{E}\Gamma_{-j} + 1} + \frac{1}{\tau + \mathbb{E}\Gamma_{-j} + 1} - \frac{1}{\tau + \mathbb{E}\Gamma + 1} \right|$$

$$\leq \sum_{j \in \mathcal{I}} 2p_j \left| \mathbb{E}(\frac{1}{\tau + \Gamma_{-j} + 1}) - \frac{1}{\tau + \mathbb{E}\Gamma_{-j} + 1} \right| + \left| \frac{1}{\tau + \mathbb{E}\Gamma_{-j} + 1} - \frac{1}{\tau + \mathbb{E}\Gamma + 1} \right|$$

$$\leq \sum_{j \in \mathcal{I}} 2p_j \left( \frac{1}{(\tau + \mathbb{E}\Gamma_{-j} + 1)^2} + \frac{p_j}{(\tau + \mathbb{E}\Gamma_{-j} + 1)(\tau + \mathbb{E}\Gamma + 1)} \right)$$

$$\leq \sum_{j \in \mathcal{I}} 2p_j \left( \frac{1}{(\tau + \mathbb{E}\Gamma)^2} + \frac{p_j}{(\tau + \mathbb{E}\Gamma)^2} \right) = \frac{\sum_{j \in \mathcal{I}} 2p_j(1 + p_j)}{(\tau + \mathbb{E}\Gamma)^2} \leq \frac{2}{\tau + \mathbb{E}\Gamma}.$$

Here, the third inequality follows from (11). The last inequality is because $\sum_{j \in \mathcal{I}} p_j \leq |\mathcal{I}| = \tau$ and $\sum_{j \in \mathcal{I}} p_j^2 \leq \sum_{j \in \mathcal{I}} p_j = \mathbb{E}\Gamma$. $\qquad \square$

# D  Training Details

The training settings mainly follow Wang et al. [2023a,b]. For Pascal VOC, Cityscapes and ADE20K, AdamW optimizer with a weight decay of 0.01 is used. The learning rate starts from $1e - 6$ and linearly warms up during the first $1\%$ iterations to the initial learning rate $6e - 5$. The learning rate is then decayed in a "poly" policy with an exponent of 1. The number of warm-up iterations is 400 for Pascal VOC and Cityscapes, and 800 for ADE20K. The total number of training iterations is 40,000 for Pascal VOC and Cityscapes, and 80,000 for ADE20K. Data augmentations including (i) random scaling in the range of $[0.5, 2.0]$, and (ii) random horizontal flipping with a probability of 0.5.

For LiTS and KiTS, we train the models using SGD with an initial learning rate of 0.01, momentum of 0.9, and weight decay of 0.0005. The learning rate is decayed in a "poly" policy with an exponent of 0.9. The batch size is 8 and the number of epochs is 60. These two datasets are originally multi-class segmentation tasks, but we convert them into binary segmentation by only treating the tumor as foreground. This is because we want to compare our method with RankDice-BA, which is only applicable to binary segmentation. Furthermore, since LiTS and KiTS do not include designated test sets, we employ 5-fold cross-validation to evaluate performance, following existing literature [Qin et al., 2021].

# E  Additional Results

## E.1  Statistical Significance Test

Table 7: Mean performance in mIoU[I] and p-values from t-tests between RankDice-RMA and Argmax-prob.

|  | Pascal VOC | Cityscapes | ADE20K |
|---|---|---|---|
| Argmax-prob | $87.80 \pm 0.12$ | $75.63 \pm 0.04$ | $56.88 \pm 0.09$ |
| RankDice-RMA | $88.16 \pm 0.11$ | $76.11 \pm 0.07$ | $57.67 \pm 0.11$ |
| p-value | 1.12e-6 | 2.30e-13 | 5.96e-13 |

To validate the statistical significance of the performance improvement achieved by RankDice-RMA over Argmax-prob, we conduct 10 independent runs with different random seeds using UPerNet on VOC, Cityscapes, and ADE20K datasets.

We report mean and standard deviation of mIoU$^I$, along with the p-values from t-tests, as shown in Table 7. The results indicate that the improvements are statistically significant, with p-values far below $0.01$ across all datasets.

More importantly, our method not only achieves a substantial improvement in the sense of mean performance, but also consistently outperforms Argmax-prob in every single run. For example, the results of the ten runs on ADE20K are presented in Table 8. This consistency arises because our method is deterministic and introduces no inherent randomness. It is applied to trained models by simply replacing Argmax-prob in the prediction step. Consequently, the comparison is highly stable as they share the same model.

Table 8: mIoU$^I$ of 10 independent runs on ADE20K.

| Run | 1 | 2 | 3 | 4 | 5 | 6 | 7 | 8 | 9 | 10 |
|---|---|---|---|---|---|---|---|---|---|---|
| Argmax-prob | 56.84 | 56.86 | 56.93 | 57.01 | 56.77 | 56.94 | 57.01 | 56.80 | 56.84 | 56.77 |
| RankDice-RMA | 57.56 | 57.66 | 57.71 | 57.86 | 57.59 | 57.73 | 57.84 | 57.59 | 57.68 | 57.52 |

## E.2 More Qualitative Visualizations

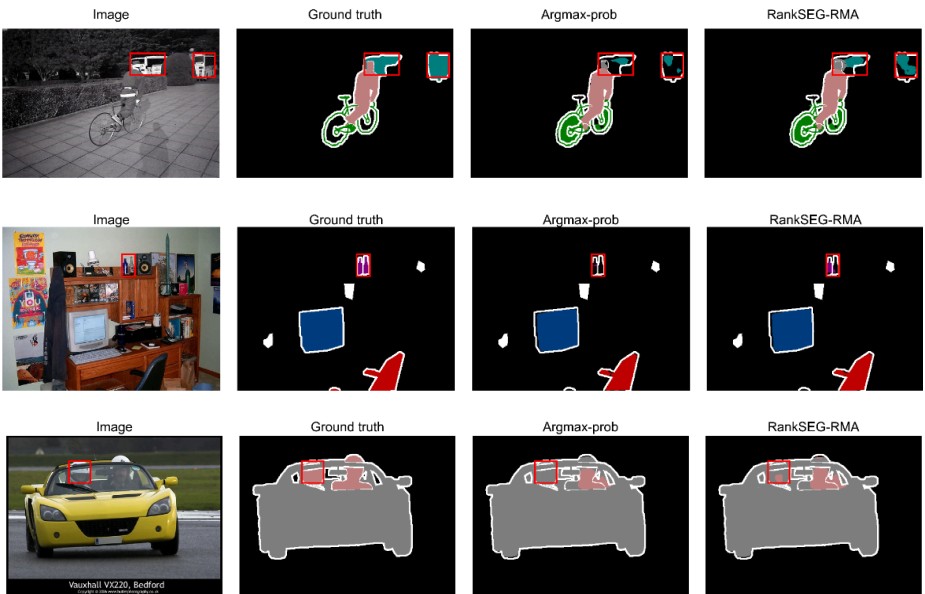

Figure 3: Qualitative visualizations on Pascal VOC. From left to right: input image, ground truth, prediction by Argmax-prob, and prediction by RankSEG-RMA. The key differences are highlighted in red boxes.

We provide additional qualitative visualizations in Figure 3 to compare the proposed RankSEG-RMA with the conventional Argmax-prob method, offering further insights into how our approach enhances segmentation quality. As highlighted by the red boxes in the figure, RankSEG-RMA outperforms Argmax-prob primarily in capturing complete regions of challenging objects and in detecting small objects.

For instance, in the first row, where the buses are partially occluded, Argmax-prob only sparsely identifies a small portion of the buses, whereas RankSEG-RMA achieves more complete segmentation. In the second example, Argmax-prob fails to detect the small bottles on the table, but RankSEG-RMA successfully identifies them. Similarly, in the third example, Argmax-prob completely misses one human face, while RankSEG-RMA detects it. These examples, together with the discussions in Sections 4.3 and 4.4, demonstrate that RankSEG-RMA is particularly effective for segmenting small and challenging objects.

