# OpenReview forum: "RankSEG-RMA: An Efficient Segmentation Algorithm via Reciprocal Moment Approximation"
_NeurIPS.cc/2025/Conference — NeurIPS 2025 poster_

### Official Review · Reviewer_fg59 · 2025-06-18

**Clarity:** 3
**Significance:** 1
**Originality:** 2
**Rating:** 5
**Confidence:** 4

**Summary:**

The paper presents a novel approach to produce semantic segmentation results, where the standard argmax operation is replaced by RankSEG-RMA, their proposed approach, which improves RankSEG. In particular, the proposed reciprocal moment approximation-based approach is significantly faster than RankSEG and is not affected by RankSEG's main limitation, namely its applicability only to overlapping segments.
The authors introduce their method with a compelling theoretical analysis and present all proofs in the appendix. Additionally, they perform extensive experiments on several datasets to prove that their proposed approach outperforms both argmax and RankSEG, while being extremely more efficient than RankSEG.

**Questions:**

* The first point of discussion is the practical applicability of the proposed method compared to existing approaches. Despite being better than argmax, the proposed approach is too slow for real-world applicability. It would be interesting to visualize some more qualitative results, showing that the proposed approach has a small but significant/critical difference in performance wrt the baselines.

* In the introduction (lines 45-55), the authors briefly discuss alternative methods used in the literature for attempting to directly optimize IoU, such as soft-IoU and Lovasz loss. Despite their theoretical inconsistencies, they are still among the most used methods for post-processing the output of a semantic segmentation network. Thus, to have a fair comparison to existing state-of-the-art techniques, it would be interesting to see a comparison against these methods as well.

* I'd suggest a minor revision of Sec. 2 and 3, as it is not clear from the text that binary segmentation is being discussed up until Sec. 3 (line 153), when it's first explicitly mentioned. In the background section in particular, concepts such as segmentation mask and predicted mask are introduced with the underlying assumption of binary segmentation, without stating it explicitly.

* In the conclusion, it is mentioned that the assumption of conditional independence in the Bayes rule is a limitation of the approach. It would be good to have an explanation of why that is, and what this assumption implies in this case.

* I suggest minor revisions of the layout to make the paper look even better than it already does. Small fixes, such as avoiding text between Algorithm 1 and Figure 1, and pushing Algorithm 2 to the top of the page rather than having text on top, would help readability and avoid losing parts of the text.

**Ethical Concerns:**

["NO or VERY MINOR ethics concerns only"]

**Final Justification:**

I want to thank the authors for their response. They carefully addressed all reviews and responded to the reviewers' doubts. For my specific case, I am satisfied with how the authors justified their choices. I also think that they did a good job on all other reviews.
The paper was already in a good shape originally, and by carefully re-reading it I am still convinced that its contribution is valuable and relevant. I encourage the authors to implement all changes and extras in the final version.
Thus, I increase my rating from a 4 to a 5.

**Limitations:**

Yes.

**Paper Formatting Concerns:**

No concern.

**Quality:**

3

**Strengths And Weaknesses:**

The paper is extremely well-written and explained, relatively easy to follow, and theoretically and technically sound. The authors do a good job of explaining the pertinent background knowledge necessary for a full understanding of the paper, and back up their claims and contributions with convincing proofs, extensive experiments, and careful and well-thought-out considerations.
The topic is also relevant and significant for the community. Despite being an extension of an existing work, the proposed approach is useful and convincing, and it's -as the authors rightfully state in the manuscript- a plug-and-play module that can be universally applied to all networks for semantic segmentation. This feature is also highlighted in the experimental evaluation, where different models are used, both CNN-based and transformer-based, showing the broad applicability of the contributed module.

The main weakness of this work is its significance in terms of practical applicability, as the performance yielded by RankSEG-RMA is slightly superior to argmax (1-1.5% mIoU in the best cases, sometimes also <1% as in Cityscapes), despite being two orders of magnitude slower on multiclass datasets. This makes the practical applicability of the method extremely limited, as it would definitely be good for benchmarking, but impossible to use for real-world deployment of any kind.

---

> ### Author Rebuttal · Authors · 2025-07-29
>
> > Q1. The main weakness of this work is its significance in terms of practical applicability, by being two orders of magnitude slower than argmax on multiclass datasets.
>
> **Reply.** Thank you for raising this important point and providing us with the opportunity to clarify. In Table 3, we separately provide the *model-forward time* and the *prediction-algorithm time*. Yet, from a practical perspective, what we truly care about is the *total segmentation time*, which is the sum of these two components.
>
> To enhance clarity, we have modified Table 3 to present the values in the format of *total time (model-forward time + prediction time)*, with "NA" indicating not applicable.
>
> |  | VOC | Cityscapes | ADE20K | LiTS | KiTS |
> | - | - | - | - | - | - |
> | Argmax-prob | 40.82 (40.77 + 0.05) | 176.30 (175.81+0.22) | 325.02 (324.59+0.43) | 14.16 (14.15+0.01) | 11.87 (11.86+0.01) |
> | RankSEG-RMA (ours) | 47.70 (40.77 + 6.93) | 185.96 (175.81+10.15) | 382.59 (324.59+58.00) | 14.49 (14.15+0.34) | 12.12 (11.86+0.26) |
> | RankSEG-BA (Dai et. al) | NA | NA | NA | 30.48 (14.15+16.33) | 21.85 (11.86+9.99) |
>
> As indicated in the table, the total time of our method is approximately 17%, 5%, and 5% greater than that of argmax on VOC, Cityscapes, and LiTS, respectively. This demonstrates that our method maintains an acceptable running time for total segmentation prediction, especially in contrast to the original RankSEG algorithm, which requires nearly twice the time.
>
> In summary, although our method is slower than argmax in terms of prediction-algorithm time, we believe that its contribution to the total segmentation time remains acceptable because it is the model-forward time overwhelmingly dominating the total time.
>
> > Q2. It would be interesting to visualize some more qualitative results, showing that the proposed approach has a small but significant/critical difference in performance wrt the baselines.
>
> **Reply.** We highly appreciate your valuable suggestion. We recognize the importance of visual comparisons in demonstrating the effectiveness of our proposed approach.
>
> We have included Fig. 2 to provide a visual comparison with the baseline. In the revised version, we will add more qualitative visualizations to further highlight the critical differences.
>
> > Q3. Despite their theoretical inconsistencies, they are still among the most used methods semantic segmentation. It would be interesting to see a comparison against soft-IoU Loss, Lovasz Loss as well.
>
> **Reply.** Thank you for your comment. We provide a comprehensive comparison with soft-IoU loss below to clarify the positioning of our work:
>
> a) **Different roles in the segmentation pipeline**: Soft-IoU losses are used during the training phase to guide model optimization, whereas our method serves a plug-and-play module that can be seamlessly integrated into a trained model at the inference stage.
>
> b) **Hyper-parameter tuning**: Perhaps due to the non-convexity of the soft-IoU losses, they are typically combined with cross-entropy loss using a weighted sum to stablize training, thus introducing an additional hyper-parameter. Tuning this parameter requires retraining the model, which is computationally expensive. In contrast, our method does not introduce any new hyper-parameters or require model retraining.
>
> c) **Probabilistic interpretation**: Producing reliable pixel-wise probabilities is crucial for uncertainty estimation [1], especially in high-stakes scenarios such as medical imaging. Models trained with soft-IoU loss do not output calibrated probabilities [1], locking their ability to quantify uncertainty. Our method, on the other hand, is typically applied to models trained with cross-entropy loss, thus preserving this probabilistic interpretation.
>
> d) **Performance comparison**: It is acknowledged that soft-IoU loss is more effective when used independently. However, since these methods operate at different stages, we can still apply RankSEG-RMA after training with soft-IoU. Specifically, we consider $L=\lambda L_{\text{CE}} + (1-\lambda) L_{\text{soft-IoU}}$ with varying $\lambda$ on ADE20k, and report the results as follows:
>
> |$\lambda$|1|0.75|0.5|0.25|0|
> |-|-|-|-|-|-|
> |Argmax|56.94|59.13|60.00|60.71|54.70|
> |RankSEG-RMA|57.67|59.50|60.17|60.81|54.68|
>
> As shown, RankSEG-RMA can still achive improvements when the model trained with the weighted loss (except for $\lambda=0$). The performance gain decreases as $\lambda$ decreases. This trend occurs because, as the soft-IoU loss becomes dominant (i.e., when $\lambda$ is small), the model outputs become less calibrated and less interpretable as probabilities, which in turn reduces the effectiveness of our method.
>
> [1] Mehrtash, Alireza, et al. "Confidence calibration and predictive uncertainty estimation for deep medical image segmentation." IEEE Transactions on Medical Imaging 39.12 (2020): 3868-3878.
>
> > Q4. I'd suggest a minor revision of Sec. 2 and 3, as it is not clear from the text that binary segmentation is being discussed up until Sec. 3. In the background section in particular, concepts such as segmentation mask and predicted mask are introduced with the underlying assumption of binary segmentation, without stating it explicitly.
>
> **Reply.** Thank you for your suggestions to enhance our work.
>
> To improve the readability, we will incorporate more explicit statements to clarify the scope of different sections. At the beginning of Section 2, we will explicitly state that the focus is on binary segmentation, and we will clarify that the notations and Bayes rule are tailored for this binary case. Furthermore, we will indicate that multi-class segmentation will be formally addressed in Section 3.2.
>
> > Q5. It is mentioned that the assumption of conditional independence in the Bayes rule is a limitation of the approach. It would be good to have an explanation of why that is, and what this assumption implies in this case.
>
> **Reply.** A limitation of the segmentation Bayes rule is that the ranking structure reflected in the final solution will be broken when Conditional Independence Assumption (CIA) is violated. When CIA is not satisfied, the Bayes rule is difficult to optimize and compute.
>
>
> Specifically, the CIA states that the label distributions of two pixels are independent given the image. Intuitively, this assumption implies that the remaining information in labels at different pixel locations are independent after removing the information from the image.
>
> Yet, we want to further explain why this assumption is still acceptable in segmentation applications. a) The CIA is **implicitly assumed when training with cross-entropy (CE) loss**, as it allows the joint likelihood to be factorized into pixel-wise terms. b) Conditional dependence is difficult to formulate practically, especially given the high dimensionality of segmentation data. c) Starting from v3 [2], DeepLab discards conditional random fields (CRFs) that were previously used to model pairwise label dependence, after developing more powerful neural network. This change suggests that **CIA is a reasonable approximation once the image ($\mathbf{X}$) delivers sufficient information (and this information is captured by powerful model) for inferring the label ($\mathbf{Y}$), the dependence among the remaining parts ($\mathbf{Y}$ after removing information from $\mathbf{X}$) becomes less significant**.
>
>
> [2] Chen, Liang-Chieh, et al. "Rethinking atrous convolution for semantic image segmentation." arXiv preprint arXiv:1706.05587 (2017).
>
> > Q6. I suggest minor revisions of the layout to make the paper look even better than it already does. Small fixes, such as avoiding text between Algorithm 1 and Figure 1, and pushing Algorithm 2 to the top of the page rather than having text on top, would help readability and avoid losing parts of the text.
>
>
> **Reply.** We highly appreciate your helpful suggestions. Thank you for your helpful suggestions. We will revise the layout as recommended to improve readability, including avoiding text between Algorithm 1 and Figure 1 and placing Algorithm 2 at the top of the page.

---

> ### Comment · Area_Chair_T7yr · 2025-08-06
> **Author-reviewer Discussion**
>
> Dear Reviewer fg59,
>
> The system noticed that you haven't post any discussion with the authors yet. As per the review guidelines, reviewers are expected respond to authors’ rebuttal, ask further questions (if any) and listen to answers to help clarify remaining issues before submitting Mandatory Acknowledgement. Please engage with the authors now to offer your feedback on their rebuttal.
>
> Thank you!
>
> AC, NeurIPS 2025

---

### Official Review · Reviewer_xrAb · 2025-06-23

**Clarity:** 3
**Significance:** 3
**Originality:** 3
**Rating:** 4
**Confidence:** 5

**Summary:**

This work introduces RankSEG-RMA, an extension of RankSEG that tackles its high computational cost and binary-class limitation by integrating Reciprocal Moment Approximation (RMA) to dramatically reduce complexity and introducing an RMA-score—an alternative to standard probability scores—for resolving overlapping regions in multi-class segmentation. Comprehensive experiments on diverse datasets and backbones demonstrate its effectiveness.

**Questions:**

- To further demonstrate the method’s generalizability, it would be beneficial to evaluate RankSEG-RMA's performance on more recent architectures, such as Mask2Former [r1] or CPT [r2]. Such experiments would provide a more robust assessment of its utility in the current state-of-the-art landscape.
- There is a minor typo on line 176. The text reads "class x," but it seems to be "class c."
- Line 69 points out that the original RankSEG is not suitable for panoptic segmentation. While the proposed RankSEG-RMA successfully extends its capability to multi-class semantic segmentation, its applicability to other segmentation tasks like panoptic or instance segmentation remains unclear.

[r1] Cheng, Bowen, et al. "Masked-attention mask transformer for universal image segmentation." Proceedings of the IEEE/CVF conference on computer vision and pattern recognition. 2022.

[r2] Tang, Quan, et al. "Rethinking Feature Reconstruction via Category Prototype in Semantic Segmentation." IEEE Transactions on Image Processing (2025).

**Ethical Concerns:**

["NO or VERY MINOR ethics concerns only"]

**Final Justification:**

The rebuttal addresses most concerns. I am maintaining the current rating at 4.

**Limitations:**

Yes

**Quality:**

3

**Strengths And Weaknesses:**

**Strengths**
- This work convincingly demonstrates that RankSEG-RMA not only enhances segmentation performance but also reduces computational complexity. The reported inference speed is substantially higher than that of the original RankSEG, marking a significant practical improvement.
- The RMA-score is effective mechanism for overlapping areas.
- The claims are well-substantiated with both theoretical derivations and comprehensive experimental analysis, lending strong credibility to the proposed method.

**Weaknesses**
- The adopted baseline, i.e., PSPNet, DeepLabV3+, SegFormer, UperNet, are actually outdated. It is strongly recommend adding one or two state-of-the-art methods to more rigorously validate the effectiveness of RankSEG-RMA.
- The work could be further strengthened by including more qualitative visualizations. Side-by-side visual comparisons with RankSEG and argmax baseline would intuitively showcase the advantages of RankSEG-RMA.
- The time consumed in multi-class semantic segmentation is still significantly greater than that of Argmax-prob as shown in Table 3.

---

> ### Author Rebuttal · Authors · 2025-07-29
>
> > Q1. The adopted baseline models are outdated. It is recommend adding one or two state-of-the-art methods to validate the effectiveness of RankSEG-RMA, for example, Mask2Former or CPT
>
> **Reply.** Thank you for your valuable suggestion. Due to time constraints and several practical challenges, we are currently unable to include results for these models at this stage. Specifically: (a) Mask2Former outputs a set of candidate masks, which is structurally different from mainstream models. Converting its output into a standard probability mask suitable for our method would require additional effort. (b) Since CPT is a very recent model, we need more time to thoroughly review their documentation and detailed implementations to ensure a fair comparison. Yet, we appreciate your suggestion and will include the references in our discussion section. We will also try out best to incorporate them in revision.
>
> Nevertheless, we believe that the models we have adopted (UNet, PSPNet, DeepLabv3+, SegFormer, UPerNet) remain among the most representative in semantic segmentation. Our primary goal is to evaluate RankSEG-RMA through a clean and fair comparison against argmax and RankSEG-BA. Furthermore, the consistent improvements demonstrated across various models and datasets suggest that our method has strong potential for generalization to other architectures.
>
> We appreciate your feedback and will strive to incorporate these state-of-the-art models in future revision.
>
> > Q2. The work could be further strengthened by including more qualitative visualizations. Side-by-side visual comparisons with RankSEG and argmax baseline would intuitively showcase the advantages of RankSEG-RMA.
>
>
> **Reply.** Thank you for your valuable suggestion. We have included Fig. 2 to provide a visual comparison with the baseline, and we will include more qualitative visualizations and comparisons in the revision to further demonstrate our work.
>
> > Q3. The time consumed in multi-class semantic segmentation is still significantly greater than that of Argmax-prob as shown in Table 3.
>
> **Reply.** Thank you for raising this important point and providing us with the opportunity to clarify. In Table 3, we separately provide the *model-forward time* and the *prediction-algorithm time*. Yet, from a practical perspective, what we truly care about is the *total segmentation time* being the sum of these two components.
>
> Thus, we have modified Table 3 for clarity as follows, presenting the values in the format of *total time (model-forward time + prediction time)*. "NA" indicates not applicable.
>
> |  | VOC | Cityscapes | ADE20K | LiTS | KiTS |
> | - | - | - | - | - | - |
> | Argmax-prob | 40.82 (40.77 + 0.05) | 176.30 (175.81+0.22) | 325.02 (324.59+0.43) | 14.16 (14.15+0.01) | 11.87 (11.86+0.01) |
> | RankSEG-RMA (ours) | 47.70 (40.77 + 6.93) | 185.96 (175.81+10.15) | 382.59 (324.59+58.00) | 14.49 (14.15+0.34) | 12.12 (11.86+0.26) |
> | RankSEG-BA (Dai et. al) | NA | NA | NA | 30.48 (14.15+16.33) | 21.85 (11.86+9.99) |
>
> As indicated in the table, the total time of our method is approximately 17%, 5%, and 5% greater than that of argmax on VOC, Cityscapes, and LiTS, respectively. This demonstrates that, when considering the total time, our method maintains the entire segmentation prediction within an acceptable running time, in contrast to the original RankSEG algorithm, which required nearly twice the time.
>
> In summary, although our method is slower than argmax in terms of prediction-algorithm time, we believe that its contribution to the total segmentation time remains acceptable because the forward pass time overwhelmingly dominates the total time.
>
> > Q4. There is a minor typo on line 176. The text reads "class x," but it seems to be "class c."
>
> **Reply.** Thank you for pointing this out. We will correct the typo in the revised version.
>
> > Q5. Its applicability to other segmentation tasks like panoptic or instance segmentation remains unclear.
>
>
> **Reply.** Thank you for bringing these questions to us. In principle, RankSEG-RMA can be applied to any segmentation task as long as a (estimated) probability mask $\mathbf{q} \in [0,1]^{C\times d}$ is provided. However, in the context of instance segmentation or panoptic segmentation, popular deep models typically produce more complex outputs; therefore, RankSEG-RMA also requires additional processing to generate the corresponding results.
>
> For example, MaskFormer [1] produces a set of $(\mathbf{p}\_i, \mathbf{m}\_i)\_{i=1}^N$ for an image, where $N$ is a pre-defined number of mask candidates, $\mathbf{p}\_i \in [0,1]^C$ denotes the class probability, and $\mathbf{m}\_i \in [0,1]^d$ denotes the binary probability mask. To adapt our method, we first broadcast $\mathbf{p}\_i$ across every pixel of $\mathbf{m}\_i$ to form a probability mask $\mathbf{q}\_i \in [0,1]^{C\times d}$ such that $q_{i, c, j}=p\_{i,c} m\_{i, j}$. We then apply RankSEG-RMA to obtain a segmentation mask. The final challenge lies in resolving overlaps between different masks. We believe this can be addressed using our RMA scores or other heuristic methods proposed by MaskFormer.
>
> We appreciate your comment and recognize the importance of exploring the potential of RankSEG-RMA in instance and panoptic segmentation tasks. We plan to investigate this further in our future work
>
> [1] Cheng, Bowen, Alex Schwing, and Alexander Kirillov. "Per-pixel classification is not all you need for semantic segmentation." Advances in Neural Information Processing Systems.

---

> > ### Comment · Reviewer_xrAb · 2025-08-05
> >
> > Thanks for the response, but core concerns remain. I insist that experiments based on the most recent SOTA will further improve this work and suggest the advancement of the method. I will wait and see other reviewers' feedback.

---

> > > ### Author Response · Authors · 2025-08-07
> > >
> > > Dear Reviewer xrAb,
> > >
> > > We appreciate the time and effort you have devoted to reviewing our work and providing valuable feedback.
> > >
> > > We have made significant efforts over the past few days to address your main concern regarding experiments on the most recent state-of-the-art segmentation models. Specifically, we have implemented the recent CPT model, following the details in the CPT's paper and the official open-sourced codebase, using the Swin-Large backbone on the ADE20K dataset. We then compared our proposed RankSEG-RMA with the Argmax approach. The results are as follows:
> > >
> > > |           | Argmax | RankSEG-RMA |
> > > |-----------|--------|-------------|
> > > | mIoUI      | 57.85  | 58.63       |
> > > | mDiceI     | 64.75  | 65.83       |
> > >
> > > As shown above, our proposed RankSEG-RMA remains effective on the CPT model. Combined with existing experimental results on other models, we reasonably conclude the effectiveness and generalization ability of our method.
> > >
> > > We hope these additional results address your primary concern. We would be grateful if you could let us know if you have any further questions and whether we have satisfactorily addressed your concern.

---

> > > > ### Comment · Reviewer_xrAb · 2025-08-07
> > > >
> > > > Thanks for your further response. After carefully reading other reviewers' feedback, I am maintaining the rating at 4 and leaning toward an acceptance. It is strongly suggested that the authors supplement these results accordingly.

---

### Official Review · Reviewer_Aq2y · 2025-06-25

**Clarity:** 3
**Significance:** 3
**Originality:** 3
**Rating:** 5
**Confidence:** 4

**Summary:**

This paper proposes RankSEG-RMA, an efficient semantic segmentation algorithm based on Reciprocal Moment Approximation (RMA). The algorithm effectively reduces the computational complexity of RankSEG from O(d log d) and O(d²) to O(d) while maintaining comparable performance and extending it to non-overlapping segmentation settings. Experiments demonstrate that RankSEG-RMA outperforms the conventional argmax rule across multiple datasets and models, with significant gains in small objects and challenging classes.

**Questions:**

please see weakness.

**Ethical Concerns:**

["NO or VERY MINOR ethics concerns only"]

**Final Justification:**

The author has addressed most of my concerns.

**Limitations:**

The paper compares RankSEG-RMA primarily with the argmax rule and RankDice-BA. However, it lacks a comprehensive comparison with other state-of-the-art segmentation methods that also aim to optimize IoU or Dice metrics. This makes it difficult to fully assess the novelty and superiority of the proposed method.

**Quality:**

3

**Strengths And Weaknesses:**

Adv:
1. The introduction of Reciprocal Moment Approximation (RMA) effectively reduces the computational complexity of RankSEG, making it more feasible for practical applications.
2. Experimental results show that RankSEG-RMA outperforms the conventional argmax rule across multiple datasets and models, particularly in small objects and challenging classes.
3. The successful extension of the RankSEG framework to non-overlapping segmentation settings provides an effective solution for standard benchmarking.

Dis:
1. The introduction of the RMA technique is somewhat complex, with detailed step-by-step explanations lacking. For example, in the proof of Theorem 2, the transition from the approximation of Poisson-binomial random variables to the final RMA formula is not intuitive for non-specialists (Section 3.1). More intermediate steps and illustrations are recommended to aid understanding.
2. While some experiments provide standard deviations, there is no statistical significance testing for performance differences between methods. For example, in Tables 1 and 2, although mIoU and mDice values are listed, it is unclear whether these differences are statistically significant (Section 4.2). This makes it difficult for readers to assess the reliability of the performance improvements.
3. What are the reasons for the different performance gains on different datasets of the model? I think a relevant clarification is in order.
4. The model has very limited enhancement on some datasets and methods, for example, only 0.4% on the DeepLabV3+ model, PASCAL dataset, which leads to some doubts about the generality of the model and the validity of the proposed methods.
5. Although the paper provides some implementation details, it lacks comprehensive information on how to reproduce the experiments, such as specific environment setups and exact commands. This could hinder the reproducibility of the results.

---

> ### Author Rebuttal · Authors · 2025-07-30
>
> > Q1. The introduction of the RMA technique is somewhat complex, with detailed step-by-step explanations lacking. For example, in the proof of Theorem 2, the transition from the approximation of Poisson-binomial random variables to the final RMA formula is not intuitive for non-specialists (Section 3.1). More intermediate steps and illustrations are recommended to aid understanding.
>
>
> **Reply.** We appreciate your feedback on this matter and will provide more detailed explanations in the revised version.
>
> a) We first clarify the motivation behind using RMA. In the original RankSEG, according to Eq (2), the computational difficulty arises from expanding the expectation $\mathbb{E}(\frac{2}{\tau + \Gamma + 1})$ with respect to $\Gamma$ into a sum of $d$ terms using the probability mass function.
>
> **Our main goal is to find a bound for $\mathbb{E}(\frac{2}{\tau + \Gamma + 1})$ that is both easy to compute and as tight as possible. To this end, RMA (Theorem 2) provides such a tight bound, and both the lower and upper bounds can be efficiently computed in $\mathcal{O}(1)$ time.**
>
> b) Next, we provide further clarification on the proof of RMA. In the proof of Theorem 2, Lemma C.1 introduces a technique to express a reciprocal moment (RM) $\mathbb{E}(\frac{1}{X+a})$ in terms of the probability generating function (PGF) $\mathbb{E}(t^X)$ for a random variable $X$. We then establish a connection between the PGF of the Poisson-binomial distribution and that of the Binomial distribution, using the arithmetic and geometric means inequality.
>
> Specifically, denoted $\Lambda(d, \bar{p})$ a Binomial with $\bar{p}=\frac{1}{d}\sum_{j=1}^dp_j$, we observe
> $$
> \mathbb{E}(t^{\Gamma}) = \prod_{j=1}^d (1-p_j+p_jt) \le (1-\bar{p}+\bar{p}t)^d = \mathbb{E}(t^{\Lambda}).
> $$
> This allows us to relate the RMs of $\Gamma$ and $\Lambda$
> $$
> \mathbb{E}(\frac{1}{\Gamma+\tau}) = \int_0^1 t^{\tau-1} \mathbb{E}(t^{\Gamma}) dt \le \int_0^1 t^{\tau-1} \mathbb{E}(t^{\Lambda}) dt = \mathbb{E}(\frac{1}{\Lambda+\tau}).
> $$
> Finally, we apply Lemma C.2, which provides a known upper bound for the reciprocal moment of a Binomial random variable, to conclude the proof.
>
> > Q2. There is no statistical significance testing for performance differences between methods. It is unclear whether these differences are statistically significant (Section 4.2)
>
>
> **Reply.** Thank you for your comments. We have now conducted 10 runs with different random seeds using UPerNet on ADE20k, Cityscapes, and VOC, and report the p-values from t-tests below.
>
> | | ADE20K | Cityscapes | VOC |
> |-|-|-|-|
> |Argmax-prob|$56.88\pm 0.09$|$75.63\pm 0.04$|$87.80 \pm 0.12$|
> |RankSEG-RMA|$57.67\pm 0.11$|$76.11\pm 0.07$|$88.16 \pm 0.11$|
> |*p-value*|5.96e-13|2.30e-13|1.12e-6|
>
> The resulting p-values indicate that the differences are statistically significant. Moreover, **our method not only achieves a substantial improvement in the sense of mean performance, but also consistently outperforms the baseline in every single run**. For example, the results of 5 out of 10 runs on ADE20K are shown below:
>
> |run_id|1|2|3|4|5|
> |-|-|-|-|-|-|
> |Argmax-prob|56.84|56.86|56.93|57.01|56.77|
> |RankSEG-RMA|57.56|57.66|57.71|57.86|57.59|
>
> Due to time and computation resource constraints, we are currently unable to perform such statistical testing across all models, but we will make every effort to include them in the revision.
>
> We would also like to note that the proposed RankSEG-RMA itself is deterministic and introduces no inherent randomness; it can be applied to any trained model by simply replacing the argmax-prob in the prediction step. As a result, the comparison between argmax-prob and our method is highly stable as they share the same trained model, which is further supported by the consistent improvements observed across different models and datasets.
>
> > Q3. What are the reasons for the different performance gains on different datasets of the model?
>
> **Reply.** Thank you for your insightful observation. RankSEG-RMA (7) is particularly effective for challenging classes (where class probabilities are close) and small objects (where region scale has a greater impact). For example, on datasets like PASCAL VOC, where the baseline (Argmax-prob) already achieves high performance (~85 mIoU) due to large object sizes, our method provides marginal improvements. In contrast, datasets such as ADE20K, which contain more small objects, and medical datasets (LiTS, KiTS) with inherent ambiguity and difficulty, demonstrate greater performance gains from RankSEG-RMA. Therefore, we believe that the inherent difficulty of the segmentation problem itself is a key factor influencing performance gains.
>
> > Q4. The model has very limited enhancement on some datasets and methods, for example, only 0.4% on the DeepLabV3+ model, PASCAL dataset
>
>
> **Reply.** Thank you for your comment. As noted in our response to your Q3, the limited improvement on some datasets is mainly because the baseline model already achieves high performance on these relatively easy datasets. In such cases, there is less room for further improvement. However, significant gains are observed on more challenging datasets, such as ADE20K and LiTS, as shown in the following table:
>
> |  | ADE20K | KiTS | LiTS |
> | - | - | - | - |
> | Argmax-pob | 56.94 | 61.16 | 38.45 |
> | Abs. Imp. | 0.73 | 2.40 | 2.26 |
> | Rel. Imp. | 1.27% | 3.92% | 5.71% |
>
> Additionally, substantial improvements are also noted in difficult classes within easier datasets, with gains of 2.56% for *Bottle* and 2.09% for *Potted Plant*, as detailed in Table 4.
>
>
> > Q5. Although the paper provides some implementation details, it lacks comprehensive information on how to reproduce the experiments, such as specific environment setups and exact commands.
>
> **Reply.** Thank you for the comment. We appreciate the importance of reproducibility and will provide additional details in Appendix D to further facilitate replication of our experiments. As noted in the manuscript, we have included an anonymized GitHub link containing the codebase necessary to reproduce our results. In particular, the main algorithm can be found in `exp/rankseg_rma.py`. We will also add more comprehensive documentation and step-by-step instructions for running the experiments, and we will release trained model checkpoints to ensure that all results can be reproduced.
>
> > Q6. It lacks a comprehensive comparison with other state-of-the-art segmentation methods that also aim to optimize IoU or Dice metrics. This makes it difficult to fully assess the novelty and superiority of the proposed method.
>
>
> **Reply.** Thank you for your comment. We believe you are referring to loss functions such as soft-Dice and soft-IoU, which are designed to directly optimize Dice or IoU metrics. We provide a comprehensive comparison below to clarify the positioning of our work:
>
> a) **Different roles in the segmentation pipeline**: Soft-Dice/IoU losses are used during the training phase to guide model optimization, whereas our method is a plug-and-play module that can be seamlessly integrated into a trained model at the inference stage.
>
> b) **Hyper-parameter tuning**: Perhaps due to the non-convexity of the Dice/IoU losses, they are typically combined with cross-entropy loss using a weighted sum, introducing an additional hyper-parameter. Tuning this parameter requires retraining the model, which is computationally expensive. In contrast, our method does not introduce any new hyper-parameters or require model retraining.
>
> c) **Probabilistic interpretation**: Producing reliable pixel-wise probabilities is crucial for uncertainty estimation [1], especially in high-stakes scenarios such as medical imaging. Models trained with soft-Dice/IoU loss do not output calibrated probabilities [1], locking their ability to quantify uncertainty. Our method is applied to models trained with cross-entropy loss, thus preserving this probabilistic interpretation.
>
> d) **Performance comparison**: It is acknowledged that soft-Dice/IoU losses are more effective when used independently. However, since these methods operate at different stages, we can still apply RankSEG-RMA after training with soft-IoU. Specifically, we consider $L=\lambda L_{\text{CE}} + (1-\lambda) L_{\text{soft-IoU}}$ with varying $\lambda$ on ADE20k, and report the results as follows:
>
> |$\lambda$|1|0.75|0.5|0.25|0|
> |-|-|-|-|-|-|
> |Argmax|56.94|59.13|60.00|60.71|54.70|
> |RankSEG-RMA|57.67|59.50|60.17|60.81|54.68|
>
> As shown, RankSEG-RMA can still achive improvements when the model trained with the weighted loss (except for $\lambda=0$). The performance gain decreases as $\lambda$ decreases. This is because, as the soft-IoU loss becomes dominant (i.e., when $\lambda$ is small), the model outputs become less calibrated and less interpretable as probabilities, which in turn reduces the effectiveness of our method.
>
> [1] Mehrtash, Alireza, et al. "Confidence calibration and predictive uncertainty estimation for deep medical image segmentation." IEEE Transactions on Medical Imaging (2020): 3868-3878.

---

> > ### Comment · Reviewer_Aq2y · 2025-08-06
> >
> > I have read the rebuttal and the comments from other reviewers. The responses have effectively addressed most of my concerns, and I am satisfied with the clarifications provided. Given the improvements and the authors’ thorough replies, I have decided to increase my score.

---

### Official Review · Reviewer_imyv · 2025-07-03

**Clarity:** 3
**Significance:** 3
**Originality:** 4
**Rating:** 4
**Confidence:** 4

**Summary:**

The paper proposes RankSEG-RMA, an improved version of the RankSEG framework for semantic segmentation.

Its core idea is to significantly reduce computational complexity via reciprocal moment approximation (RMA), while preserving theoretical consistency in optimizing Dice/IoU metrics. It also introduces a pixel-wise scoring function for non-overlapping multiclass segmentation.
Experimental validation is conducted on natural image datasets (Cityscapes, PASCAL VOC, ADE20K) and medical image datasets (LiTS, KiTS). Compared to the traditional argmax scheme, the proposed method achieves modest improvements in average IoU/Dice and worst-case quantile performance.

**Questions:**

Please refer to the weaknesses and those are my main concerns. I have two minor questions and these questions would not influence my score:

1. Could RankSEG-RMA be extended to instance segmentation or panoptic segmentation tasks?
2. Can the authors quantify Dice/IoU discrepancies caused by the RMA approximation compared to original RankSEG? Could certain extreme probability distributions (for example, long-tail) negatively impact segmentation quality due to approximation?

**Overall, if I misunderstand some details about this paper, please let me know.** I am looking forward to discuss with authors in rebuttal phase.

**Ethical Concerns:**

["NO or VERY MINOR ethics concerns only"]

**Final Justification:**

I think the response has resolved most of concerns. Considering the reviews from other reviewers, rebuttal discussions among reviewers and authors, I would keep my initial score and lean to an acceptance for this paper. Authors should update their paper according to these reviews.

**Limitations:**

yes

**Quality:**

3

**Strengths And Weaknesses:**

Overall, I think this paper demonstrates a high degree of originality, is supported by solid theoretical analysis, and holds strong practical value. It brings a new way for adapting RankSEG into semantic segmentation with less computation time.

## Strengths

1. The paper builds on the theoretical foundation of RankSEG, inheriting and strengthening theoretical consistency with Dice/IoU metrics. By employing a Bayesian segmentation rule, the authors ensure convergence to the optimal Dice/IoU solution in infinite-data scenarios, a guarantee that standard cross-entropy + argmax frameworks cannot offer. Compared to surrogate-loss-based empirical optimization, this method is more theoretically sound, highlighting high originality and quality.

2. The authors cleverly propose Reciprocal Moment Approximation (RMA) to approximate IoU/Dice expectations, reducing algorithmic complexity from O(d^2) to O(d). I think this results in significant speedups (dozens of times) without considerable loss in accuracy.

3. The paper addresses RankSEG’s original limitation of only supporting overlapping labels by introducing a pixel-level scoring function for non-overlapping multiclass segmentation. This strategy makes the method directly applicable to mainstream semantic segmentation benchmarks, which broadens the applicability compared to the original RankSEG.

4. As RankSEG-RMA only modifies the inference step without altering model training, it can easily integrate into existing segmentation model deployments as a performance-enhancing post-processing module. This plug-and-play nature is highly appealing in industrial applications, highlighting the method’s potential significance in practice. In the experiments, the authors also demonstrate this by adopting to different methods.


## Weaknesses

Despite the above strengths, the following issues/weaknesses/concerns should be taken seriously by the authors, and I'm looking forward to the response from authors:

1. Although theoretically optimal, actual empirical improvements in average metrics are relatively modest (generally within 1%).
For instance, the mIoU improvement on Cityscapes is less than 1%, with some metrics even slightly decreasing in certain datasets.cImprovements in worst-case quantiles are also modest (around 1%). So, I wonder that whether the complexity introduced by the method is justified by the incremental gains achieved. Such limited improvement may not always be sufficient to researchers to replace existing solutions, somewhat weakening practical significance.


2. In my opinion, the theoretical optimality proof relies on a crucial assumption: **pixel labels being conditionally independent given the image**. However, adjacent pixels typically exhibit strong correlation (region coherence in segmentation segmentation tasks). This independence assumption simplifies theoretical derivation but is unlikely to hold in real scenarios, thus it is only approximately optimal I think. In this paper, the authors do not discuss how violating this assumption might affect method performance (Emmmm, maybe they discussed, so please point out if I missing this).

3.  The introduced RMA-score for non-overlapping segmentation is essentially heuristic and does not guarantee globally optimal multiclass segmentation. Pixels are simply assigned to the highest-scoring class, a strategy close to argmax probability with adjusted scores. Thus, my concern is that In some extreme cases (e.g., very close probabilities between two classes with complex global IoU trade-offs), this greedy approach might fail to deliver globally optimal Dice/IoU. After reviewing this paper, I think it lacks rigorous analysis of the optimality or approximation ratio in multiclass scenarios, only providing empirical evidence of effectiveness.

4. I also think there maybe a fairness issue. While performance on small objects improves, optimizing IoU pixel-wise inherently biases toward majority classes or large regions. RankSEG-RMA itself does not explicitly address class imbalance problem (long-tail problem)
For rare classes (such as some classes in LVIS), ranking optimization might be limited by scarce positive pixels.


5. Some equtions are hard to understanding. I try my best to understand the formulas and the reasoning behind the derivations. Therefore, I suggest that the authors refine the presentation to improve clarity. Making the paper more accessible would broaden its impact and reduce the barrier to understanding.

---

> ### Author Rebuttal · Authors · 2025-07-30
>
> >Q1.1 Although theoretically optimal, empirical improvements are relatively modest.
>
> **Reply.** Thank you for your comment. We summarize the absolute and relative gains in mIoU (with strongest backbones for argmax to avoid overstating improvements):
>
> | |VOC|Cityscapes|ADE20K|KiTS|LiTS
> -|-|-|-|-|-
> argmax|87.82|75.66|56.94|61.16|38.45
> Abs. Imp.|0.53|0.51|0.73|2.40|2.26
> Rel. Imp.|0.60%|0.67%|1.27%|3.92%|5.71%
>
> While improvements are modest on easier datasets (VOC, Cityscapes: <1%), larger gains are obseved on more challenging datasets (ADE20K: 1.3%, KiTS: 3.9%, LiTS: 5.7%). Furthermore, improvements for difficult or small classes are substantial—for example, in VOC, *Bottle*, *Chair*, and *Potted Plant* improve by 2.56%, 2.47%, and 2.09% (see Table 4).
>
> In summary, while the gains on easier datasets may be modest, they are noteworthy for harder datasets and challenging classes within easier datasets.
>
> >Q1.2 With some metrics slightly decreasing in certain datasets.
>
> **Reply.** Perhaps we misunderstood, yet Tables 1 and 2 show consistent gains across datasets. We would greatly appreciate it if you could point out the specific case.
>
> >Q1.3 Whether the complexity introduced is justified by the incremental gains.
>
> **Reply.** We would like to clarify that the **complexity introduced is minimal** for two reasons: a) The plug-and-play nature of RankSEG-RMA requires only minor implementation effort to deployment; b) The overall segmentation time increases by less than 10% on average, which is acceptable in practice.
>
> Given these relatively low overheads and the improvements on challenging datasets and classes, we believe RankSEG-RMA provides a reasonable practical contribution.
>
> >Q2. The theoretical optimality proof relies on conditionally independence assumption.
>
> **Reply.** We agree that the conditional independence assumption (CIA) may be violated in some real datasets. We argue this assumption is still acceptable by following points:
>
> a) **Already used in training**: CIA is **implicitly assumed when training with cross-entropy (CE) loss**, as it allows the joint likelihood to be factorized into pixel-wise terms. Without this assumption, the likelihood or CE formulation would be computationally infeasible. Specifically, the decomposed negative log-likelihood is:
>
> $$
> L_n(\mathbf{q})=-\log(\prod_{i=1}^n\mathbb{P}_{\mathbf{q}}(\mathbf{Y}=\mathbf{y}_i|\mathbf{X}=\mathbf{x}_i))=-\log(\prod\_{i=1}^n \prod\_{j=1}^dq_j(\mathbf{x}_i)^{y\_{ij}}(1-q\_j(\mathbf{x}_i))^{1-y\_{ij}}) =\sum\_{i=1}^n\ell\_{\text{CE}} (\mathbf{y}_i,\mathbf{q}(\mathbf{x}_i)),
> $$
>
> where the second equality uses CIA, $\mathbf{q}$ denotes the model, and $\{(\mathbf{x}_i, \mathbf{y}_i)\}\_{i=1}^n$ is the training set. Thus, we does not introduce an additional assumption beyond standard setup.
>
> b) **Computational challenges**: Conditional dependence is hard to formulate practically, especially given the high dimensionality of segmentation. For example, even modeling pairwise dependence in a 512×512 image requires handling $512^4$ pairs, making computation infeasible.
>
> c) **Empirical precedent**: As seen in prior work, such as DeepLab v1/v2, modeling pairwise dependence with Conditional Random Fields was eventually abandoned in v3 [1] due to high computational cost and limited gains.
>
> In summary, **CIA is a reasonable approximation once the image ($\mathbf{X}$) delivers sufficient information for inferring $\mathbf{Y}$, as the residual dependence in $\mathbf{Y}$ (after removing information from $\mathbf{X}$) becomes negligible**. Therefore, CIA remains a common and practical assumption due to its computational advantages. We agree that exploring local dependence is an interesting direction for future work.
>
> [1] Chen, et al. "Rethinking atrous convolution for semantic image segmentation." ArXiv (2017).
>
> >Q3. About the heuristic and theoretical optimality of RMA-score.
>
> **Reply.** We agree RMA-score for non-overlapping is heuristic. We would like to offer following justifications for its use: a) As shown in Lemma 7 of [2], solving the Bayes rule under non-overlapping constraints is even harder than a nonlinear assignment problem (typically cubic complexity [3]), making it computationally infeasible and thus justifying the use of heuristics; b) The RMA-score has an intuitive interpretation as the expected increment in Dice/IoU according to Eq (7), providing a meaningful optimization criterion; c) It outperforms other heuristic alternatives such as probability or volume-weighted probability scores (see Sec 4.5), with gains of 0.43 and 0.39.
>
> Therefore, RMA-score offers a practical balance between theoretical motivation and computational efficiency, making it an effective choice for addressing overlaps.
>
> [2] Dai and Li. "Rankseg: a consistent ranking-based framework for segmentation." JMLR (2023)
>
> [3] Kuhn. "The Hungarian method for assignment problem." NRLQ (1955)
>
> >Q4. While performance on small objects improves, optimizing IoU pixel-wise biases toward large classes. RankSEG-RMA does not explicitly address class imbalance. For rare classes (in LVIS), ranking optimization might be limited by scarce positive pixels.
>
> **Reply.** Thank you for raising this important point. If we understand correctly, your concern relates to class size imbalance within an image (e.g., a small motorcycle versus a large car in a city scene).
>
> While RMA-score optimizes pixel-wise as in Eq (6) and Eq (7), it differs from argmax by considering the region size through the denominator terms $|\widehat{\mathcal{I}}\_c|$ and $\widehat{\mu}\_c$. This results in a score discount for larger regions, leading to **favor smaller regions** when appropriate. For example, even if $\hat{p}\_{c_1,j} > \hat{p}\_{c_2,j}$, our method could still assign $c_2$ to pixel $j$ if $\hat{\mu}\_{c\_1}< \hat{\mu}\_{c\_2}$. In contrast, argmax depends only on probability values and lacks this adaptivity.
>
> Numerically, our method also achieves greater gains on small classes, such as *Bottle* (2.56%) and *Chair* (2.47%), which occupy the fewest pixels in VOC. Therefore, we believe that for small-sized classes, RankSEG-RMA provides an improvement over argmax.
>
> >Q5. Some equations are hard to understand. I suggest refine the presentation to improve clarity.
>
> **Reply.** Thank you for your suggestion. We will add more explanations in the revision for clarity. Specifically, in Eq (2), the second equality expands the expectation $\mathbb{E}(\frac{2}{\tau + \Gamma + 1})$ into a sum of $d$ terms. This highlights the computational difficulty in the original RankSEG, and **motivates us to find an approximation for $\mathbb{E}(\frac{2}{\tau + \Gamma + 1})$ that is both easy to compute and as accurate as possible.**
>
> RMA (Theorem 2) addresses this by providing tight lower and upper bounds, both computable in $\mathcal{O}(1)$. Specifically, from Eq (2) to Eq (4), where we apply RMA, the only change is replacing $\mathbb{E}(\frac{2}{\tau +\Gamma + 1})$ with $\frac{2}{\tau + \mathbb{E}(\Gamma)+1}$. This change reduces the expectation to a linear function of $\Gamma$, rather than a non-linear one, yielding significant computational gains.
>
> >Q6. Could RankSEG-RMA be extended to instance/panoptic segmentation?
>
> **Reply.** Thank you for your question. In principle, RankSEG-RMA can be applied to any segmentation task given a (estimated) probability mask $\mathbf{q} \in [0,1]^{C\times d}$. However, for instance or panoptic segmentation, where models often produce more complex outputs, additional processing would be needed.
>
> For example, MaskFormer [4] produces a set of $(\mathbf{p}_i, \mathbf{m}_i)\_{i=1}^N$ for an image, where $N$ is the number of mask candidates, $\mathbf{p}_i \in [0,1]^C$ denotes class probability, and $\mathbf{m}_i \in [0,1]^d$ denotes binary probability mask. To adapt our method, we could broadcast $\mathbf{p}_i$ across every pixel of $\mathbf{m}_i$ to form a probability mask $\mathbf{q}_i \in [0,1]^{C\times d}$ with $q\_{i, c, j}=p\_{i,c}m\_{i, j}$, and then apply RankSEG-RMA to obtain a segmentation mask. Overlaps between masks can potentially be resolved using our RMA scores or other heuristic methods in MaskFormer. We appreciate your question and will explore this potential in future work.
>
> [4] Cheng, et. al "Per-pixel classification is not all you need for semantic segmentation." Neurips (2021)
>
> >Q7.1 Can you quantify discrepancies caused by the RMA compared to original RankSEG?
>
> **Reply.** We quantify the quality of the RMA both theoretically and numerically. Theoretically, Lemma 3 provides a theoretical error bound, which is tight for large mask volumes. Numerically, our method and original RankSEG yield nearly identical results (63.56 vs. 63.56 on KiTS; 40.71 vs. 40.70 on LiTS), confirming RMA’s practical accuracy. We will add more results and discussion in Table 2.
>
> For multi-class segmentation, such quantification is not feasible since original RankSEG is not readily applicable.
>
> >Q7.2 Could extreme probability distributions (e.g., long-tail) negatively impact the quality due to approximation?
>
> **Reply.** To assess the impact of extreme distributions, we simulated a 5-class segmentation in a square grid, where pixels closer to the four corners and center are more likely to belong to classes 1-5. The logits at pixel $(h,w)$ for $c=1,\cdots,5$ are:
> $$\text{logits}(h,w)=\exp(-d_c(h,w)/(H+W)),$$
>
> where $d_c$ denotes the distance to the four-corners (e.g., left-top) and the center. The exponential decay yields long-tailed logits. We then apply $\text{softmax}(\text{logits}/t)$ to obtain probabilities, where $t$ controls the long-tailness. **Larger $t$ implies a more uniform distribution, while smaller $t$ produces a more pronounced long tail by concentrating probability mass on top classes.**
>
> With $H=W=32$ and a sample size of 3000, the results below show that our method remains effective under long-tailed distributions:
>
> t|0.7|0.8|0.9|1
> -|-|-|-|-
> argmax|13.78|13.32|12.96|12.63
> ours|14.49|14.09|13.74|13.43

---

> > ### Comment · Reviewer_imyv · 2025-08-06
> >
> > Thanks for the authors' reply. I think the response has resolved most of concerns. Considering the reviews from other reviewers, rebuttal discussions among reviewers and authors, I would keep my initial score and lean to an acceptance for this paper. Authors should update their paper according to these reviews.

---

> ### Comment · Area_Chair_T7yr · 2025-08-06
> **Author-reviewer Discussion**
>
> Dear Reviewer imyv,
>
> The system noticed that you haven't post any discussion with the authors yet. As per the review guidelines, reviewers are expected respond to authors’ rebuttal, ask further questions (if any) and listen to answers to help clarify remaining issues before submitting Mandatory Acknowledgement. Please engage with the authors now to offer your feedback on their rebuttal.
>
> Thank you!
>
> AC, NeurIPS 2025

---

### Comment · Area_Chair_T7yr · 2025-08-03
**Reviewer-Author Discussion**

Dear Reviewers,

The discussion period with the authors will remain open until August 6th (AoE). Please take the time to read and acknowledge the authors' rebuttals, and post any follow-up questions or comments you may have.

Best regards, AC

---

### Note · Authors · 2025-08-12

We sincerely appreciate the AC for taking the lead in coordinating this review process, and thank all reviewers for their dedication during both the review and rebuttal stages. We are grateful for their acknowledgment of our strengths and their constructive suggestions for improving our work.

In this rebuttal, we have made every effort to clarify and address the following concerns raised by the reviewers:

1. **Modest Improvement**: While we recognize the modest improvements (<1%) on easier datasets like VOC, we highlight significant gains on more challenging datasets (ADE20K: 1.3%, KiTS: 3.9%, LiTS: 5.7%). Notably, improvements in difficult or small classes are substantial, which is particularly relevant in contexts such as medical imaging.

2. **Time Consumption**: We clarify that our method requires only a minor increase in total segmentation time (~10%), in a sharp contrast to the original RankSEG, which demands twice the time for binary segmentation and is computationally infeasible for multi-class cases.

3. **Conditional Independence Assumption**: We provide a thorough discussion of this assumption, showing that it is a reasonable approximation and is also used implicitly in existing methods.

4. **Statistical Significance**: We present results from 10 runs with different random seeds, confirming that our improvements are statistically significant. Importantly, gains are observed in every run, as our method does not introduce any randomness and shares the same trained model for comparisons with the argmax.

5. **Application to Recent Models**: We have additionally implemented one of the most recent models, CPT, and demonstrated that our method remains effective.

6. **Comparison to Soft-IoU Loss**: We provide a comprehensive comparison with soft-IoU loss to better position our work.

We are pleased that several reviewers found our responses satisfactory. We will incorporate all these discussions and results into our revision. We again thank the AC and all reviewers for their invaluable time and engagement.

---

### Decision · Program_Chairs · 2025-09-17

**Decision:**

Accept (poster)

**Comment:**

This submission proposes RankSEG-RMA, an extension of RankSEG for semantic segmentation. The key idea is to employ a Reciprocal Moment Approximation to reduce computational complexity while maintaining consistency with Dice/IoU metrics. All the reviewers rated positively for this paper, praising its theoretical soundness, computational efficiency, plug-and-play nature, and extensive experiments. However, there are also concerns regarding the magnitude of improvements, runtime overhead, and evaluation scopes.

The authors engaged extensively with the reviewers. Their rebuttal clarified the runtime overhead (arguing that the additional inference cost is minor compared to forward pass), addressed theoretical assumptions, and provided new experiments, including statistical significance tests and results on CPT. These responses satisfied most reviewers. Reviewer's concerns about outdated baselines and practical applicability remain partly unresolved but were mitigated by the explanations.

In general, this paper makes a theoretically well-founded and practically useful contribution to segmentation tasks, and all the reviewers aggree on acceptance. There is no special reason to overturn the concensus.